# DIFFUSION-NESTED AUTO-REGRESSIVE SYNTHESIS OF HETEROGENEOUS TABULAR DATA

## ABSTRACT

Autoregressive models are predominant in natural language generation, while their application in tabular data remains underexplored. We posit that this can be attributed to two factors: 1) tabular data contains heterogeneous data type, while the autoregressive model is primarily designed to model discrete-valued data; 2) tabular data is column permutation-invariant, requiring a generation model to generate columns in arbitrary order. This paper proposes a Diffusion-nested Autoregressive model (TABDAR) to address these issues. To enable autoregressive methods for continuous columns, TABDAR employs a diffusion model to parameterize the conditional distribution of continuous features. To ensure arbitrary generation order, TABDAR resorts to masked transformers with bi-directional attention, which simulate various permutations of column order, hence enabling it to learn the conditional distribution of a target column given an arbitrary combination of other columns. These designs enable TABDAR to not only freely handle heterogeneous tabular data but also support convenient and flexible unconditional/conditional sampling. We conduct extensive experiments on ten datasets with distinct properties, and the proposed TABDAR outperforms previous state-of-the-art methods by 18% to 45% on eight metrics across three distinct aspects.

## 1 INTRODUCTION

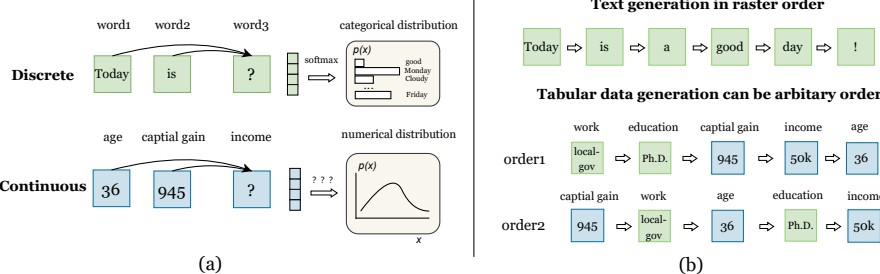

Figure 1: Challenges in Auto-Regressive tabular data generation. (a) The conditional distribution of continuous columns is hard to express. (b) Tabular data is column-permutation-invariant.

Due to the widespread application of synthetic tabular data in real-world scenarios, such as data augmentation, privacy protection, and missing value prediction (Fonseca & Bacao, 2023; Assefa et al., 2021; Hernandez et al., 2022), an increasing number of studies have begun to focus on deep generative models for synthetic tabular data generation. In this domain, various approaches, including Variational Autoencoders (VAEs)(Liu et al., 2023), Generative Adversarial Networks (GANs)(Xu et al., 2019), Diffusion Models (Zhang et al., 2024b), and even Large Language Models (LLMs)(Borisov et al., 2023), have demonstrated significant progress. However, Auto-Regressive models, a crucial category of generative models, have been largely overlooked in this process. In language modeling, autoregressive models have become the *de facto* solution (e.g., GPTs (Mann et al., 2020; Achiam et al., 2023)). Tabular data shares similarities with natural language in its discrete structure, making an autoregressive decomposition of its distribution a natural approach, i.e.,

$$p(\mathbf{x}) = p(x^1) \prod_{i=2}^{D} p(x^i | x^1, x^2, \cdots, x^{i-1}) = p(x^1) \prod_{i=2}^{D} p(x^i | \mathbf{x}^{<i}) \tag{1}$$

where each $x^i$ represents the value at the $i$-th column, $\mathbf{x}^{<i} = \{x^1, x^2, \cdots, x^{i-1}\}$. However, research on autoregressive models for tabular data generation has not received adequate attention.

We posit that this is primarily due to two key challenges (see Fig. 1): 1) **Modeling continuous distribution**: Autoregressive models aim to learn the per-token (conditional) probability distribution, which is convenient for discrete tokens (such as words in Natural Languages) because their probability can be represented as a categorical distribution. However, the same approach is challenging to apply to continuous tokens unless there are strong prior assumptions about their distribution, such as assuming they follow Gaussian distributions. 2) **No fixed order**: Unlike natural language, which possesses an inherent causal order from left to right, tabular data exhibits column permutation invariance . To reflect this property in the autoregressive model, we need to ensure a sequence of tokens can be generated in arbitrary order. Existing autoregressive models for tabular data generation adopt simplistic approaches to address these challenges. For instance, they discretize continuous columns, allowing them to learn corresponding categorical distributions (Castellon et al., 2023; Gulati & Roysdon, 2023). However, this method inevitably leads to information loss. Regarding the generation order, these models typically default to a left-to-right column sequence (Castellon et al., 2023), failing to reflect the column permutation invariant property.

To address these challenges, this paper proposes **D**iffusion-nested **A**uto**R**egressive **Tab**ular Data Generation (TABDAR in short). TABDAR addresses the aforementioned issues through two design features: 1) **Nested diffusion models** for modeling the conditional probability distributions of the next continuous-valued tokens. TABDAR nests a small diffusion model (Ho et al., 2020; Karras et al., 2022) into the autoregressive framework for learning the conditional distribution of a continuous-valued column. Specifically, we employ two distinct loss functions for learning the distribution of the next continuous/discrete-valued tokens, respectively. For discrete columns, their conditional distribution is learned by directly minimizing the KL divergence between the prediction vector and the ground-truth one-hot category embedding. For continuous columns, we learn their distribution through a diffusion model conditioned on the output of the current location. In this way, TABDAR can flexibly learn the distribution of tabular data containing arbitrary data types. 2) **Masked Transformers with bi-directional attention**. TABDAR simulates arbitrary sequence orders via a transformer-architectured model with bi-directional attention mechanisms and masked inputs. The masked/unmasked locations indicate the columns that are missing/observed, ensuring that when predicting a new token, the model has only access to the known tokens. During training, TABDAR follows the form of masked language modeling, predicting the distribution of masked tokens based on unmasked tokens. During testing, TABDAR generates entire rows of data in an autoregressive manner according to a given order.

TABDAR offers several advantages. 1) TABDAR employs the most appropriate generative models for different data types, i.e., diffusion models for continuous columns and categorical prediction for discrete columns, seamlessly integrating them into a unified framework. This avoids contrived processing methods such as discrete diffusion models (Lee et al., 2023; Kotelnikov et al., 2023), continuous tokenization of categorical columns (Zhang et al., 2024b), and ineffective discretization (Castellon et al., 2023). 2) Through an autoregressive approach, TABDAR models the conditional distribution between different columns, thereby better capturing the dependency relationships among various columns and achieving superior density estimation. 3) Combining autoregression and masked Transformers enables a trained TABDAR to compute and sample from the conditional distribution of target columns given an arbitrary set of observable columns, enabling generation in arbitrary order. This capability facilitates precise and convenient posterior inference (e.g., conditional sampling and missing data imputation).

We conduct comprehensive experiments on ten tabular datasets of various data types and scales to verify the efficacy of the proposed TABDAR. Experimental results comprehensively demonstrate TABDAR's superior performance in: **1) Statistical Fidelity**: The ability of synthetic data to faithfully recover the ground-truth data distribution; **2) Data Utility**: The performance of synthetic data in downstream Machine Learning tasks, such as Machine Learning Efficiency; and **3) Privacy Protection**: Whether the synthetic data is sampled from the underlying distribution of the training data rather than being a simple copy. In missing value imputation tasks, TABDAR exhibits remarkable performance, even surpassing state-of-the-art methods that are specially designed for missing data imputation tasks. For a genuine and fair comparison, we have released the code to reproduce our method and all baseline results. The code is available at `https://anonymous.4open.science/r/ICLR-TabDAR`.

## 2 RELATED WORKS

**Synthetic Tabular Data Generation**    Generative models for tabular data have become increasingly important and have widespread applications Assefa et al. (2021); Zheng & Charoenphakdee (2022); Hernandez et al. (2022). For example, CTGAN and TAVE (Xu et al., 2019) deal with mixed-type tabular data generation using the basic GAN (Goodfellow et al., 2014) and VAE (Kingma & Welling, 2013) framework. GOGGLE (Liu et al., 2023) incorporates Graph Attention Networks in a VAE framework such that the correlation between different data columns can be explicitly learned. DP-TBART (Castellon et al., 2023) and TabMT (Gulati & Roysdon, 2023) use discretization techniques to numerical columns and then apply autoregressive transformers for a generation. Recently, inspired by the success of Diffusion models in image generation, a lot of diffusion-based methods have been proposed, such as TabDDPM (Kotelnikov et al., 2023), STaSy (Kim et al., 2023), CoDi (Lee et al., 2023), and TabSyn (Zhang et al., 2024b), which have achieved SOTA synthesis quality.

**Autoregressive Models for Continuous Space Data**    In text generation, autoregressive next-token generation is undoubtedly the dominant approach (Mann et al., 2020; Achiam et al., 2023). However, in image generation, although autoregressive models (Van den Oord et al., 2016; Salimans et al., 2017) were proposed early on, their pixel-level characteristics limited their further development. Subsequently, diffusion models, which are naturally suited to modeling continuous distributions, have become the most popular method in the field of image generation. In recent years, some studies have attempted to use discrete-value image tokens (van den Oord et al., 2017; Razavi et al., 2019) and employ autoregressive transformers for image-generation tasks (Kolesnikov et al., 2022). However, discrete tokenizers are both difficult to train and inevitably cause information loss. To this end, recent work has attempted to combine continuous space diffusion models with autoregressive methods. For example, Li et al. (2024b) employs an autoregressive diffusion loss in a causal Transformer for learning image representations; Li et al. (2024a) proposes using a diffusion model to model the conditional distribution of the next continuous image and employs a masked bidirectional attention mechanism to enable the generation of any number of tokens in arbitrary order.

## 3 PRELIMINARIES

### 3.1 DEFINITIONS AND NOTATIONS

In this paper, we always use uppercase boldface (e.g., $\mathbf{X}$) letters to represent matrices, lowercase boldface letters (e.g., $\mathbf{x}$) to represent vectors, and regular italics (e.g., $x$) to denote scalar entries in matrices or vectors. Tabular data refers to data organized in a tabular format consisting of rows and columns. Each row represents an instance or observation, while each column represents a feature or variable. In this work, we consider heterogeneous tabular data that may contain both numerical and categorical columns or only one of these types. Let $\mathcal{D} = \{\mathbf{x}_i\}_{i=1}^{N}$ denote a tabular dataset comprising $N$ instances, where each instance $\mathbf{x} = (x^1, x^2, \ldots, x^D)$ is a $D$-dimensional vector representing the values of $D$ features or variables. We further categorize the features into two types: 1) Numerical/continuous features: $\mathcal{N} = \{i \mid x^i \in \mathbb{R}\}$ is the set of indices corresponding to numerical features. 2) Categorical/discrete features: $\mathcal{C} = \{i \mid x^i \in \mathcal{C}_i\}$ is the set of indices corresponding to categorical features, where $\mathcal{C}_i$ is the set of possible categories for the $i$-th feature. Note that $\mathcal{N} \cup \mathcal{C} = 1, 2, \ldots, D$ and $\mathcal{N} \cap \mathcal{C} = \emptyset$.

### 3.2 DIFFUSION MODELS

Diffusion models (Ho et al., 2020; Song et al., 2021; Karras et al., 2022) learn the data distribution $p(\mathbf{x})$ through a diffusion SDE, which consists of a forward process that gradually adds Gaussian noises of increasing scales to $\mathbf{x}$ (which are pre-normalized to have zero-mean and unit-variance), and a reverse process that recovers the clean data from the noisy one[1]:

$$\mathbf{x}_t = \mathbf{x}_0 + \sigma(t)\boldsymbol{\varepsilon}, \ \boldsymbol{\varepsilon} \sim \mathcal{N}(\mathbf{0}, \mathbf{I}), \quad \text{(Forward Process)} \quad (2)$$

$$\mathrm{d}\mathbf{x}_t = -2\dot{\sigma}(t)\sigma(t)\nabla_{\mathbf{x}_t} \log p(\mathbf{x}_t)\mathrm{d}t + \sqrt{2\dot{\sigma}(t)\sigma(t)}\mathrm{d}\boldsymbol{\omega}_t, \quad \text{(Reverse Process)} \quad (3)$$

---

[1]This formulation is a simplified version of VE-SDE (Song et al., 2021). See Appendix B for details.

where $\omega_t$ is the standard Wiener process. $\sigma(t)$ is the noise schedule, and $\dot{\sigma}(t)$ is the derivative of $\sigma(t)$ w.r.t. $t$ A diffusion model is learned by using a denoising/score network $\epsilon_\theta(\mathbf{x}_t, t)$ to approximate the conditional score function $\nabla_{\mathbf{x}_t} \log p(\mathbf{x}_t|\mathbf{x}_0)$ (named score-matching). The final loss function could be reduced to a simple formulation where the denoising network is optimized to approximate the added noise $\varepsilon$, i.e.,

$$\mathcal{L}(\mathbf{x}) = \mathbb{E}_{t \sim p(t)} \mathbb{E}_{\varepsilon \sim \mathcal{N}(\mathbf{0}, \mathbf{I})} \|\epsilon_\theta(\mathbf{x}_t, t) - \varepsilon\|^2. \tag{4}$$

The sampling process starts from a large timestep $T$ such that $\boldsymbol{x}_T \approx \mathcal{N}(\mathbf{0}, \sigma^2(T)\mathbf{I})$ recovers $\mathbf{x}_0$ via solving the reverse SDE in Eq. 3, using mature numerical solution tools.

## 4 METHODS

### 4.1 KEY INGREDIENTS OF TABDAR

**Modeling tabular data using autoregression.** TABDAR follows the autoregressive criteria for modeling the distribution of tabular data. Autoregressive modeling decomposes the joint data distribution into the product of a series of conditional distributions in raster order, and the optimization is achieved by minimizing the standard negative log-likelihood:

$$p(\mathbf{x}) = \prod_{i=1}^{D} p(x^i|\mathbf{x}^{<i}), \ \mathcal{L} = -\log p(\mathbf{x}) = -\sum_{i=1}^{D} \log p(x^i|\mathbf{x}^{<i}) \tag{5}$$

Therefore, instead of directly modeling the complicated joint distribution, one can seek to model each conditional distribution, i.e., $p(x^i|\mathbf{x}^{<i})$, separately, which is intuitively much simpler.

Autoregression is natural for language modeling, as language inherently possesses a left-to-right (L2R in short) order, and we can expect that the generation of a new word depends entirely on the observed words preceding it. Therefore, Transformers (Vaswani et al., 2017) with *causal attention* (where only the previous tokens are observable to predict the following tokens) are widely used for text generation. Unlike text data, tabular data has long been considered column permutation invariant. In this case, one has to randomly shuffle the columns many times and then apply causal attention to each generated order, which is conceptually complicated.

**Simulate arbitrary order using masked Bi-directional Attention.** To deal with this, an interesting observation is that even if given the default order of data, we can simulate arbitrary-order causal attention using masked bi-directional attention. As illustrated in Fig. 2, giving the tabular data arranged in the default order ['age', 'capital gain', 'work', 'education', 'income'] and a shuffled generation order ['education' → 'income' → 'capital gain', → 'age', → 'work'], the prediction of 'capital gain' can be equivalently achieved by 1) causal attention based on the previous columns 'education' and 'income'; 2) bidirectional attention where all other columns ('age', 'capital gain', and 'work') are masked.

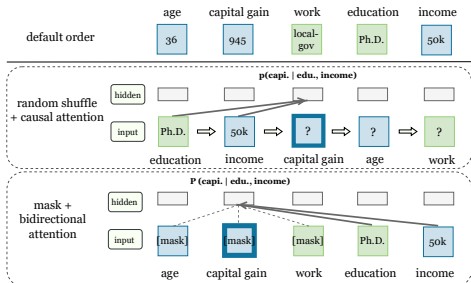

Figure 2: With appropriate masking, bidirectional attention is equivalent to causal attention in arbitrary order.

**Column-specific losses for discrete/continuous columns** Autoregressive models aim at learning the conditional distribution of the target column given the observations of previous columns, i.e., $p(x^i|\mathbf{x}^{<i})$. Take $\mathbf{x}^{<i}$ as input, Transformers are able to generate column-specific output vectors, i.e., $\mathbf{z}^i$ for column $i$. Then we only need to model the conditional distribution $p(x^i|\mathbf{z}^i)$.

*Discrete Columns.* For a discrete column $x^i$, the target distribution is a categorical distribution, which could be represented by a $|\mathcal{C}_i|$-dimensional vector. Consequently, we can directly project $\mathbf{z}^i$ to a $|\mathcal{C}_i|$-way classifier using a prediction head $f_i(\cdot)$ (e.g., a shallow MLP: $\mathbb{R}^d \to \mathbb{R}^{|\mathcal{C}_i|}$), and then minimize the KL-divergence between the predicted distribution and the one-hot encoding of a sample:

$$\mathcal{L}(x^i, \mathbf{z}^i) = -\log p(x^i|\mathbf{z}^i) = \text{Cross-Entropy}(x^i, \text{softmax}(f_i(\mathbf{z}^i)), \ i \in \mathcal{C} \tag{6}$$

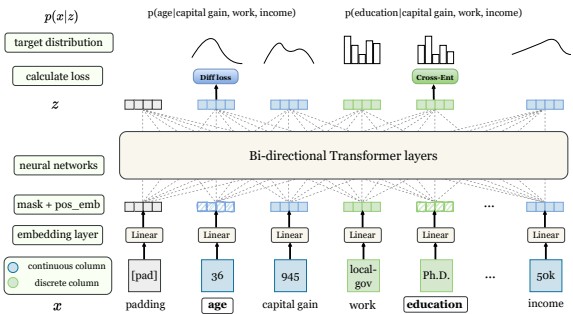
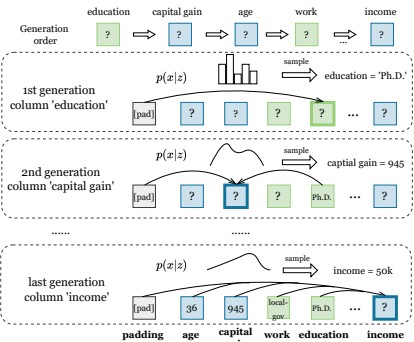

Figure 3: Framework of TABDAR. An embedding layer first encodes each column into a vector. The masks are then added to the target columns ('age' and 'education'). With Bi-direction Transformers' decoding, the output vectors $z$ are used as conditions for predicting the distribution of current columns. TABDAR nests a diffusion model in the autoregressive framework to learn the conditional distribution of a continuous column.

Figure 4: An illustration of TABDAR's generation process. Given a random generation order, e.g., 'capital gain' → 'education' → $\cdots$ 'income', TABDAR generates the value for each column in a row according to the conditional distribution learned by the masked Transformers.

*Continuous Columns.* Unlike discrete columns, a continuous column might have infinite value states and, therefore, cannot be modeled by a categorical distribution[2]. Inspired by the prominent capacity of Diffusion models in modeling arbitrary continuous distributions, we hereby adopt a conditional diffusion model to learn the conditional continuous distribution (Li et al., 2024a) $p(x^i|\mathbf{z}^i)$. Compared with the unconditional diffusion loss in Eq. 4, the denoising function $\epsilon_\theta$ takes the condition vector $\mathbf{z}^i$ as an additional input. The final loss function is expressed as follows:

$$\mathcal{L}(x^i, \mathbf{z}^i) = -\log p(x^i|\mathbf{z}^i) = -\log \mathbb{E}_{t\sim p(t)}\mathbb{E}_{\boldsymbol{\varepsilon}\sim\mathcal{N}(\mathbf{0},\mathbf{I})}\|\epsilon_\theta(x_t^i, t, \mathbf{z}^i) - \boldsymbol{\varepsilon}\|^2, \ \ i \in \mathcal{N} \tag{7}$$

### 4.2 DETAILED ARCHITECTURE OF TABDAR

In this section, we introduce the detailed architecture and implementations of TABDAR. The overall framework of TABDAR is presented in Fig. 3.

**Tokenization layer.** Given a row of tabular data $\mathbf{x} = (x^1, x^2, \cdots, x^D)$, the tokenization layer embeds each feature to a $d$-dimensional vector using column-specific learnable linear transformation. For a continuous column, the linear transformation is $\mathbf{W}^i \in \mathbb{R}^{1\times d}$, while for a discrete column, the transformation matrix is $\mathbf{W}^i \in \mathbb{R}^{|\mathcal{C}_i|\times d}$. The output token of column $i$ is denoted by $\mathbf{h}^i$, and we obtain an embedding matrix of $D$ tokens $\mathbf{H} \in \mathbb{R}^{D\times d}$. We also add column-specific learnable positional encoding so that the token embeddings at different locations can be discriminated.

**Input masking.** As illustrated in Section 4.1, we can use masked bidirectional attention to simulate arbitrary causal attention. Therefore, we first uniformly sample the number of tokens to mask/predict $M \sim \mathcal{U}(1, D)$. Given $M$, we randomly sample a masking vector $\mathbf{m} \in \{0,1\}^D$, s.t., $\sum_i m^i = M$. $m^i = 1$ indicates that column $i$ is masked and vice versa. Then, the input embeddings of masked columns are set as zero, making them unobservable when predicting the target columns.

$$\mathbf{H}' = (1 - \mathbf{m}) \odot \mathbf{H}. \tag{8}$$

**Bi-directional Transformers.** Given the masked token sequence, we then follow the standard pipeline of Transformer implementations. We first pad the embedding of the class token [pad] at the beginning of the sequence. The [pad] tokens not only enable class-conditional generation but also help stabilize the training process (Li et al., 2024a). By default, we use one unique [pad] token, equivalent to unconditional generation (we may use class-specific [pad] tokens for class-conditional distribution). Afterward, these tokens are processed by a series of standard Transformer blocks and will finally output column-wise latent vectors $\{\mathbf{z}^i\}_{i=1}^D$.

---

[2]MSE loss is also not acceptable since it is a deterministic mapping rather than a conditional distribution

**Loss function and model training.** After obtaining the latent vector $\mathbf{z}^i$ for a target/masked column $i$, we use it as the condition for predicting the distribution of $\mathbf{x}^i$, i,e., $p(x^i|\mathbf{z}^i)$.

*Discrete columns.* As presented in Eq. 6, we use a column-specific prediction head $f_i(\cdot)$ to obtain the $|\mathcal{C}_i|$-dimensional prediction then compute the cross-entropy loss.

*Continuous columns.* For predicting the distribution of the continuous column $x^i$, we use the conditional diffusion loss described in Eq. 7. The training of the conditional diffusion model follows the standard process of diffusion models (Karras et al., 2022). To be specific, we first randomly sample a timestep $t \sim p(t)$, and standard Gaussian noise $\varepsilon \sim \mathcal{N}(0, 1)$, obtaining $x_t^i = x^i + \sigma(t) \cdot \varepsilon$. Then, a denoising neural network $\epsilon_\theta(x_t^i, t, \mathbf{z}^i)$ takes $x_t^i, t$, and $\mathbf{z}^i$ as input to predict the added noise $\varepsilon$ (via the MSE loss in Eq. 7). The denoising neural network is implemented as a shallow MLP, and its detailed architecture is presented in Appendix C.

**Model Training.** The training of TABDAR is end-to-end: all the model parameters, including the Embedding layer, transformers, prediction heads for discrete columns, and diffusion models for the continuous columns, are jointly trained via gradient descent on the summation of the losses of the target/masked columns in the current batch.

**Model inference.** After TABDAR is trained, we can easily perform a variety of inference tasks, e.g., unconditional and conditional generation. An illustration of TABDAR's auto-regressive sampling process is presented in Fig. 4.

*Unconditional Sampling.* To generate unconditional data examples $\mathbf{x} \sim p(\mathbf{x})$, we can first randomly sample a generation order. Then, starting with all columns masked (except the [pad] token), we can generate the value of each column one by one according to the sampled order.

*Conditional Sampling.* One important advantage of TABDAR's autoregressive generation manner is that one can perform flexible conditional sampling, such as simple class-conditional generation. This can be achieved by directly setting the corresponding column values to the desired ones and then randomly sampling the generation order of other columns. **Missing value imputation** can be regarded as a special case of conditional sampling by resorting to the conditional distribution of missing columns given observed values. More implementation details can be found in Appendix D.4.

## 5 EXPERIMENTS

### 5.1 EXPERIMENTAL SETUPS

**Datasets**. We select ten real-world tabular datasets of varying data types and sizes: 1) two contain only continuous features – **California** and **Letter**; 2) two contain only categorical features – **Car** and **Nursery**; 3) six datasets of mixed continuous and discrete features – **Adult**, **Default**, **Shoppers**, **Magic**, **News**, and **Beijing**. The detailed introduction of these datasets can be found Appendix D.2.

**Baselines**. We compare the proposed TABDAR with ten powerful synthetic tabular data generation methods belonging to six categories. 1) The non-parametric interpolation method SMOTE (Chawla et al., 2002). 2) VAE-based methods TVAE (Xu et al., 2019) and GOGGLE (Liu et al., 2023). 3) GAN-based method CTGAN (Xu et al., 2019). 4) LLM-based method GReaT (Borisov et al., 2023). 5) Diffusion-based methods: STaSy (Kim et al., 2023), CoDi (Lee et al., 2023), TabDDPM (Kotelnikov et al., 2023), and TabSyn (Zhang et al., 2024b). 6) Autoregressive methods DP-TBART (Castellon et al., 2023) and Tab-MT (Gulati & Roysdon, 2023). The proposed TABDAR belongs to the autoregressive family, yet it utilizes a diffusion model for modeling continuous columns.

**Evaluation Methods**. We evaluate the quality of synthetic tabular data from three distinct dimensions: 1) *Fidelity* - if the synthetic data faithfully recovers the ground-truth data distribution. 2) *Utility* - the performance when applied to downstream tasks, and we focus on Machine Learning Efficiency (MLE). 3) *Privacy* - if the synthetic data is not copied from the real records. More introduction of these metrics is in Appendix D.6.

**Implementations.** Since TABDAR's diffusion loss is applied to each single column, the denoising network $\varepsilon_\theta$ can be light-weighted, and we implement it as a three-layer MLP with 256 hidden dimension. For the diffusion model, we follow Zhang et al. (2024b) and set $\sigma(t) = t$, which enables a small number of diffusion step, i.e., NFE = 50.

Table 1: Comparison of different methods regarding the **statistical fidelity** of the synthetic data. All metrics have been scaled so that lower numbers indicate better performance, to facilitate better numerical comparison.

| Method | Marginal↓ % | Joint↓ % | $\alpha$-Precision↓ % | $\beta$-Recall↓ % | C2ST↓ % | JSD↓ $10^{-2}$ |
|---|---|---|---|---|---|---|
| *Interpolation* | | | | | | |
| SMOTE (Chawla et al., 2002) | $1.72_{\pm1.36}$ | $2.95_{\pm1.66}$ | $3.78_{\pm3.94}$ | $\mathbf{16.7_{\pm9.16}}$ | $3.00_{\pm3.66}$ | $0.11_{\pm0.10}$ |
| *VAE-based* | | | | | | |
| TVAE (Xu et al., 2019) | $15.8_{\pm17.1}$ | $17.4_{\pm18.3}$ | $18.2_{\pm20.1}$ | $70.9_{\pm26.3}$ | $43.9_{\pm22.7}$ | $1.01_{\pm0.70}$ |
| GOGGLE (Liu et al., 2023) | $17.2_{\pm6.28}$ | $29.1_{\pm11.8}$ | $21.8_{\pm17.3}$ | $90.8_{\pm5.64}$ | - | - |
| *GAN-based* | | | | | | |
| CTGAN (Xu et al., 2019) | $17.9_{\pm6.99}$ | $18.4_{\pm9.11}$ | $17.7_{\pm15.1}$ | $69.1_{\pm33.8}$ | $53.0_{\pm22.5}$ | $1.18_{\pm0.69}$ |
| *LLM-based* | | | | | | |
| GReaT (Borisov et al., 2023) | $12.9_{\pm6.05}$ | $44.3_{\pm27.3}$ | $17.2_{\pm12.8}$ | $53.2_{\pm26.0}$ | $42.4_{\pm19.2}$ | $1.43_{\pm1.18}$ |
| *Diffusion-based* | | | | | | |
| STaSy (Kim et al., 2023) | $14.3_{\pm7.40}$ | $13.5_{\pm9.76}$ | $21.8_{\pm24.7}$ | $55.6_{\pm29.6}$ | $53.9_{\pm16.6}$ | $1.25_{\pm1.13}$ |
| CoDi (Lee et al., 2023) | $17.4_{\pm11.3}$ | $15.2_{\pm19.8}$ | $10.0_{\pm5.93}$ | $51.7_{\pm31.1'}$ | $44.0_{\pm33.3}$ | $0.76_{\pm0.50}$ |
| TabDDPM (Kotelnikov et al., 2023) | $15.0_{\pm25.3}$ | $7.92_{\pm8.16}$ | $23.6_{\pm2.93}$ | $49.6_{\pm34.5}$ | $24.6_{\pm38.9}$ | $1.03_{\pm1.60}$ |
| TabSyn (Zhang et al., 2024b) | $1.73_{\pm0.76}$ | $2.53_{\pm1.45}$ | $2.52_{\pm2.93}$ | $44.1_{\pm24.5}$ | $2.76_{\pm2.19}$ | $.12_{\pm0.09}$ |
| *Autoregressive-based* | | | | | | |
| DP-TBART (Castellon et al., 2023) | $3.24_{\pm1.76}$ | $2.71_{\pm1.56}$ | $2.11_{\pm2.15}$ | $48.0_{\pm27.8}$ | $5.36_{\pm4.58}$ | $0.16_{\pm0.11}$ |
| Tab-MT (Gulati & Roysdon, 2023) | $14.9_{\pm15.1}$ | $8.11_{\pm10.8}$ | $23.2_{\pm31.9}$ | $63.5_{\pm38.3}$ | $48.6_{\pm47.7}$ | $0.60_{\pm0.83}$ |
| TABDAR (ours) | $\mathbf{1.21_{\pm0.51}}$ | $\mathbf{1.80_{\pm0.72}}$ | $\mathbf{1.33_{\pm1.10}}$ | $37.2_{\pm20.1}$ | $\mathbf{1.50_{\pm1.42}}$ | $\mathbf{0.09_{\pm0.048}}$ |
| Improvement | $\mathbf{29.65\%}$ | $\mathbf{28.85\%}$ | $\mathbf{36.97\%}$ | - | $\mathbf{45.65\%}$ | $\mathbf{18.18\%}$ |

## 5.2 MAIN RESULTS

**Statistical Fidelity**   We first investigate whether synthetic data can faithfully reproduce the distribution of the original data. We use various statistical metrics to reflect the degree to which synthetic data estimates the distribution density of the original data. These metrics include univariate density estimation (i.e., the **marginal** distribution of a single column), the correlation between any two variables (which reflects the **joint** probability distribution), $\alpha$-Precision (reflecting whether a synthetic data example is close to the true distribution), $\beta$-Recall (reflecting the coverage of synthetic data over the original data), Classifier Two Sample Test (C2ST, reflecting the difficulty in distinguishing between synthetic data and real data), and Jensen-Shannon divergence (JSD, estimating the distance between the distributions of real data and synthetic data). Due to space limitations, in this section, we only present the average performance with standard deviation on each metric across all ten datasets. The detailed performance on each individual dataset is in Appendix E.

In Table 1, we present the performance comparison on these fidelity metrics. As demonstrated, our model achieved performance far surpassing the second-best method in five out of six fidelity metrics, with advantages ranging from $18.18\%$ to $45.65\%$. Considering that these metrics have already been elevated to a considerably high level due to the explosive development of recent deep generative models for tabular data, our improvement is very significant. The only exception is $\beta$-Recall, which measures the degree of coverage of the synthetic data over the entire data distribution. On this metric, the simple classical interpolation method SMOTE achieved the best performance, far surpassing other generative models. On the other hand, SMOTE also achieves very good performance on other metrics, surpassing many generative methods. These phenomena indicate that simple interpolation methods can indeed obtain synthetic data with a distribution close to that of real data. However, the limitation of interpolation methods lies in the fact that the synthetic data is too close to the real data, making it resemble a copy from the training set rather than a sample from the underlying distribution, which may cause privacy issues. Detailed experiments are in the Privacy Protection section.

**Utility on Downstream Tasks**   We then evaluate the quality of synthetic data by assessing their performance in Machine Learning Efficiency (MLE) tasks. Following previous settings (Zhang et al., 2024b), we first split a real table into a real training and a real testing set. The generative models are trained on the real training set, from which a synthetic set of equivalent size is sampled. This synthetic data is then used to train a classification/regression model (XGBoost Classifier and XGBoost Regressor (Chen & Guestrin, 2016)), which will be evaluated using the real testing set. The performance of MLE is measured by the AUC score for classification tasks and RMSE for regression tasks. As demonstrated in Table 2, the proposed TABDAR gives a fairly satisfying performance on the MLE tasks, and the classification/regression performance obtained via training on the synthetic

Table 2: AUC (classification task) and RMSE (regression task) scores of Machine Learning Efficiency. ↑ (↓) indicates that the higher (lower) the score, the better the performance.

| Method | Continuous only | | Discrete only | | Heterogeneous | | | | | |
|---|---|---|---|---|---|---|---|---|---|---|
| | California AUC↑ | Letter AUC↑ | Car AUC↑ | Nursery AUC↑ | Adult AUC↑ | Default AUC↑ | Shoppers AUC↑ | Magic AUC↑ | News RMSE↓ | Beijing RMSE↓ |
| Real data | 0.999 | 0.989 | 0.999 | 1.000 | 0.927 | 0.770 | 0.926 | 0.946 | 0.842 | 0.423 |
| *VAE-based* | | | | | | | | | | |
| TVAE | 0.986 | 0.989 | 0.746 | 0.939 | 0.846 | 0.744 | 0.898 | 0.912 | 0.979 | 1.010 |
| GOGGLE | - | - | - | - | 0.778 | 0.584 | 0.658 | 0.654 | 1.09 | 0.877 |
| *GAN-based* | | | | | | | | | | |
| CTGAN | 0.925 | 0.729 | 0.899 | 1.000 | 0.874 | 0.736 | 0.868 | 0.874 | 0.845 | 1.065 |
| *LLM-based* | | | | | | | | | | |
| GReaT | 0.996 | 0.983 | 0.979 | 0.999 | 0.913 | 0.755 | 0.902 | 0.888 | - | 0.653 |
| *Diffusion-based* | | | | | | | | | | |
| STaSy | **0.997** | 0.990 | 0.927 | 0.982 | 0.903 | 0.749 | 0.909 | 0.923 | 0.933 | 0.672 |
| CoDi | 0.981 | 0.998 | 0.995 | 1.000 | 0.829 | 0.497 | 0.855 | 0.930 | 0.999 | 0.750 |
| TabDDPM | 0.992 | 0.513 | 0.995 | 1.000 | 0.911 | 0.763 | 0.915 | 0.933 | - | 2.665 |
| TabSyn | 0.993 | 0.990 | 0.971 | 0.997 | 0.904 | 0.764 | 0.913 | 0.934 | 0.862 | 0.669 |
| *Autoregressive* | | | | | | | | | | |
| DP-TBART | 0.993 | 0.985 | 0.990 | 0.917 | 0.918 | 0.717 | 0.896 | 0.924 | 0.896 | 0.676 |
| Tab-MT | 0.988 | 0.985 | 0.981 | 1.000 | 0.873 | 0.714 | 0.912 | 0.822 | 1.002 | 2.098 |
| TABDAR | 0.994 | 0.994 | 0.996 | 1.000 | 0.904 | 0.764 | 0.916 | 0.935 | 0.856 | 0.579 |

Table 3: Probability that a synthetic example's DCR to the training set rather than that of the holdout set), a score closer to 50% is better.

| Method | Default | Shoppers |
|---|---|---|
| SMOTE | 91.41% | 96.40% |
| TabDDPM | 51.30% | 51.74% |
| TabSyn | 50.88% | 51.50% |
| TABDAR | 51.13% | 50.97% |

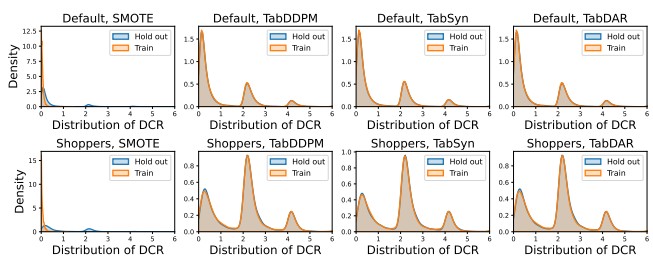

Figure 5: Distributions of the DCR scores between the synthetic dataset and the training/holdout datasets.

set is rather close to that on the original training set. Also, similar to the observations in (Zhang et al., 2024b), the differences between various methods on MLE tasks are very small, even though some methods may not correctly learn the distribution of the original data. Therefore, it is important to combine this with the fidelity metrics in Table 1 to obtain a more comprehensive evaluation of synthetic data.

**Privacy Protection** Finally, a good synthetic dataset should not only faithfully reproduce the original data distribution but also ensure that it is sampled from the underlying distribution of the real data rather than being a copy. To this end, we use the Distance to Closest Record (DCR) score for measurement. Specifically, we divide the real data equally into two parts of the same size, namely the training set and the holdout set. We train the model based on the training set and obtain the sampled synthetic dataset. Afterward, we calculate the distribution of distances between each synthetic data example and its closest sample in both the training set and the holdout set. Intuitively, if the training set and holdout set are both drawn uniformly from the same distribution and if the synthetic data has learned the true distribution, then on average, the proportion of synthetic data samples that are closer to the training set should be the same as those closer to the holdout set (both 50%). Conversely, if the synthetic data is copied from the training set, the probability of it being closer to samples in the training set would far exceed 50%.

In Table 3 and Figure 5, we present the probability comparison and the distributions of the DCR scores of SMOTE, TabDDPM, TabSyn, and the proposed TABDAR. As demonstrated, SMOTE is poor at privacy protection, as its synthetic sample tends to be closer to the training set rather than the holdout set, as its synthetic examples are obtained via interpolation between training examples. By contrast, the remaining deep generative methods are all good at preserving the privacy of training data, leading to almost completely overlapped DCR distributions.

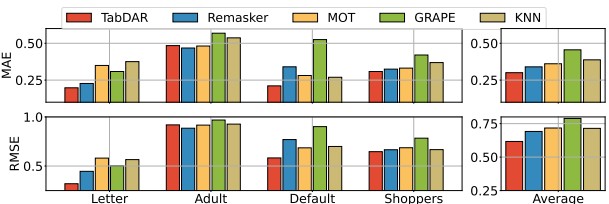

Figure 6: (30% MCAR) Comparison of missing value imputation performance with 4 competitive baselines.

Table 4: Ablation Studies: Effects of the Diffusion loss and random order sampling.

| Variants | Margin | Joint |
|---|---|---|
| w/o. both | 14.82% | 8.94% |
| w/o. Diff. loss | 12.35% | 6.11% |
| w/o. rand. order | 1.83% | 2.05% |
| TABDAR | 1.21% | 1.80% |

Table 5: Impacts of the model depth (number of Transformer layers) on Adult and Beijing.

| Depth | Margin | Joint | $\alpha$-Precision | $\beta$-Recall |
|---|---|---|---|---|
| 2 | 1.10% | 2.37% | 0.56% | 47.56% |
| 4 | 1.02% | 2.19% | 1.56% | 47.37% |
| 6 | 0.79% | 1.81% | 0.50% | 45.88% |
| 8 | 0.96% | 2.09% | 1.02% | 46.84% |

Table 6: Impacts of the embedding dimension $d$ on Adult and Beijing.

| Dim | Margin | Joint | $\alpha$-Precision | $\beta$-Recall |
|---|---|---|---|---|
| 8 | 1.15% | 2.88% | 1.15% | 53.37% |
| 16 | 0.99% | 2.25% | 0.68% | 50.27% |
| 32 | 0.79% | 1.81% | 0.50% | 45.88% |
| 64 | 1.05% | 2.24% | 1.62% | 42.21% |

## 5.3 MISSING DATA IMPUTATION

Since TABDAR directly models the factorized conditional probability of the target column(s) given the observation of other columns, it has a strong potential for missing data imputation. In this section, we compare the proposed TABDAR with the current state-of-the-art (SOTA) methods for missing data imputation, including KNN (Pujianto et al., 2019), GRAPE (You et al., 2020), MOT (Muzellec et al., 2020), and Remakser (Du et al., 2024). We consider the task of training the model on the complete training data and then imputing the missing values of testing data. The missing mechanism is Missing Completely at Random (MCAR), and the missing ratio of testing data is set at 30%. In Figure 6, we present the performance comparison on four datasets: Letter, Adult, Default, and Shoppers (as well as the average imputation performance). The proposed TABDAR demonstrates superior performance on these four datasets, significantly outperforming the current best methods on three out of four datasets. On the Adult dataset, it was slightly inferior. These results confirm that TABDAR is not only suitable for unconditional generation but also applicable to other conditional generation tasks, demonstrating a wide range of application values.

## 5.4 ABLATION STUDIES

**Effects of diffusion loss and random ordering.** We first study if the two key ingredients of TABDAR– 1) the diffusion loss and 2) random ordering – are indeed beneficial. We consider three variants of TABDAR: 'w/o. Diff. Loss', which indicates that we discretize the continuous columns into 100 uniform bins, treating them as discrete ones, such that we can apply the cross-entropy loss; 'w/o rand. order,' which indicates that we follow the default left-to-right order in both the training and generation phases using a causal attention Transformer model; 'w/o. both', which indicates the version that combines the two variants. In Table 4, we compare TABDAR with the three variants on the average synthetic data's fidelity across all the datasets. We can observe that both the diffusion loss and random training/generation order are important to TABDAR's success. Specifically, the diffusion loss targeting the modeling of the conditional distribution of the continuous columns contributes the most, and the random ordering further improves TABDAR's performance. Furthermore, random ordering enables TABDAR to perform flexible conditional inference tasks like imputation.

**Sensitivity to hyperparameters.** We then study TABDAR's sensitivity to its hyperparameters that specify its transformer architecture: the model depth (number of Transformer layers) and the embedding dimension $d$. From Table 5 and Table 6 (grey cells indicate the default setting), we can observe that although there exists an optimal hyperparameter setting (i.e., depth = 6, $d$ = 32), the change of the two hyperparameters has little impact on the model performance. These results demonstrate that our model is relatively robust to these hyperparameters. Therefore, good results can be obtained for different datasets without the need for specific hyperparameter tuning.

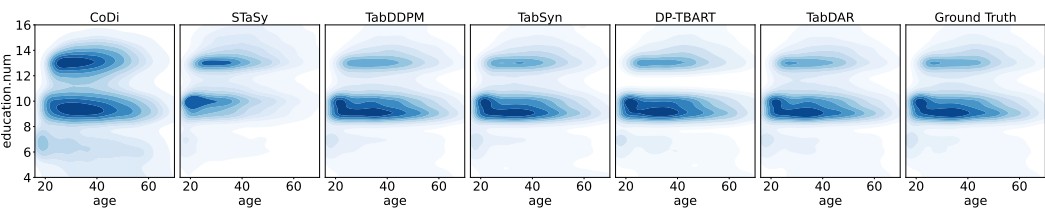

Figure 7: 2D joint distribution density of 'age' and 'education num' of Adult dataset.

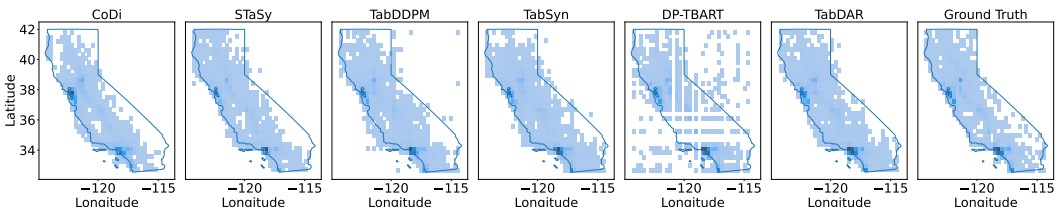

Figure 8: 2D joint distribution density of 'Longitude' and 'Latitude' of California dataset.

# 6 VISUALIZATIONS

In this section, we visualize the synthetic data to demonstrate that the proposed TABDAR can generate synthetic data that closely resembles the ground-truth distribution. In Figure 7 and Figure 8, we plot the 2D joint distribution of two columns of the Adult and California datasets to investigate if the ground-truth joint distribution density can be learned by the synthetic data. In Figure 9, we further plot the heatmaps of the estimation error of column pair correlations. These results visually demonstrate that TABDAR can generate synthetic samples very close to the distribution of real data and faithfully reflect the correlations between different columns of the data.

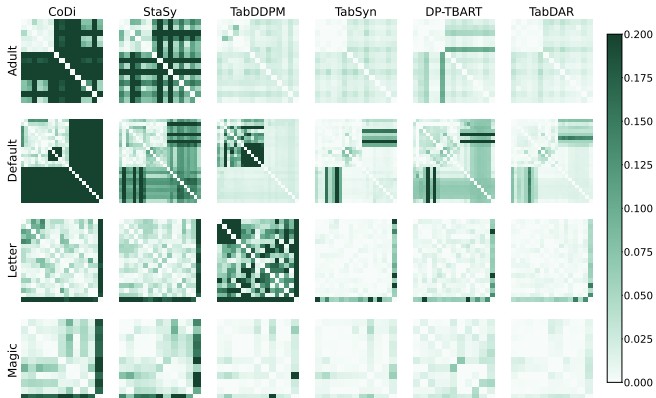

Figure 9: Heatmaps of the joint column correlation estimation error. Lighter areas indicate a lower error in estimating the correlation between two columns.

# 7 CONCLUSION

This paper has presented TABDAR, a generative model that embeds a diffusion model within an autoregressive transformer framework used for multi-modal tabular data synthesis. TABDAR uses a novel diffusion loss and traditional cross-entropy loss to learn the conditional distributions of continuous columns and discrete columns, respectively, enabling a single autoregressive model to generate both continuous and discrete features simultaneously. Furthermore, TABDAR employs masked bidirectional attention to simulate arbitrary autoregressive orders, allowing the model to generate in any direction. This not only enhances the accuracy of joint probability modeling but also enables more flexible conditional generation. Extensive experimental results demonstrate the effectiveness of the proposed method.

## REPRODUCIBILITY STATEMENT

We describe the algorithm of TABDAR in Appendix A. The detailed implementations are provided in Appendix C and Appendix D. The codes for implementing baselines and the proposed TAB-DAR are provided in the anonymous github repo https://anonymous.4open.science/r/ICLR-TabDAR.

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

# A  ALGORITHMS

In this section, we provide detailed algorithms of TABDAR. Algorithm 3 provides the training algorithm of TABDAR. Algorithm 4 and 5 provide the unconditional and conditional generation algorithm of TABDAR, respectively. For simplicity, the algorithms are presented with a single data as input, it is straightforward to extend to batch setting.

---

**Algorithm 1:** Loss for continuous columns

---

1: **Input:** Condition vector $\mathbf{z}^i$, target continuous value $x^i$, denoising network $\epsilon_\theta$
2: **Output:** Loss $\mathcal{L}(p(x^i|\mathbf{z}^i))$
3: Sample $t \sim p(t)$
4: Sample $\varepsilon \sim \mathcal{N}(0,1)$
5: Get $x_t^i = x^i + \sigma(t) \cdot \varepsilon$
6: Compute loss: $\mathcal{L}(p(x^i|\mathbf{z}^i)) = \|\epsilon_\theta(x_t^i, t, \mathbf{z}^i) - \varepsilon\|_2^2$

---

**Algorithm 2:** Loss for discrete columns

---

1: **Input:** Condition vector $\mathbf{z}^i$, target discrete value $x^i$, prediction head $f_i(\theta)$
2: **Output:** Loss $\mathcal{L}(p(x^i|\mathbf{z}^i))$
3: Get $\hat{\mathbf{x}}^i = f_i(\mathbf{z}^i)$
4: Compute loss: $\mathcal{L}(p(x^i|\mathbf{z}^i)) = \text{CrossEntropy}(x^i, \text{Softmax}(\hat{x}^i))$

---

**Algorithm 3:** TABDAR: Training

---

1: **Input**: data $\mathbf{x} = (x^1, x^2, \cdots x^D)$
2: **Output**: Model parameters
3: **1. Tokenization**
4: **for** $i \in 1, 2, \cdots, D$ **do**
5:     $\mathbf{h}^i = \mathbf{W}^i x^i$
6: **end for**
7: Let $\mathbf{H} = [\mathbf{h}^1, \mathbf{h}^2, \cdots, \mathbf{h}^D]$
8: Add positional encoding
9:
10: **2. Adding mask**
11: Sample the masking number $M \sim \text{Uniform}(1, D)$
12: Sample the masking vector $\boldsymbol{m} \in \{0,1\}^D, s.t., \sum_i m^i = M$
13: Let $\mathbf{H}' = (1 - \mathbf{m}) \odot \mathbf{H}$
14:
15: **3. Transformer layers**
16: Append padding token $[\texttt{pad}]$
17: $\mathbf{Z} = \text{Transformers}(\mathbf{H}')$
18:
19: **4. Compute Losses**
20: **for** $i \in 1, 2, \cdots, D$ **do**
21:     **if** $m^i == 1$ **then**
22:         **if** $i \in \mathcal{C}$ **then**
23:             Compute $\mathcal{L}(p(x^i|\mathbf{z}^i))$ as discrete columns ;                    ▷ Algorithm 2
24:         **else**
25:             Compute $\mathcal{L}(p(x^i|\mathbf{z}^i))$ as continuous columns ;                    ▷ Algorithm 1
26:         **end if**
27:     **end if**
28: **end for**
29: Compute $\mathcal{L} = \sum_{i=1}^{D} \mathcal{L}(p(x^i|\mathbf{z}^i)) \cdot m^i$ ;            ▷ Compute losses only for target/masked columns
30: Back-propagation to optimize model parameters

---

---

**Algorithm 4:** TABDAR: Unconditional Generation

1: **Input:** A generation order $\mathbf{o} = [o_1, o_2, \cdots, o_D] = \text{Shuffle}([1, 2, 3, \cdots D])$
2: **Output:** A sample $\tilde{\mathbf{x}} \in \mathbb{R}^D$
3: $\mathbf{m} \leftarrow \mathbf{1} \in \mathbb{R}^D$ ;                                    ▷ All tokens are unknown initially
4: $\mathbf{H} = \mathbf{0}$
5: **for** $i \in 1, 2, \cdots, D$ **do**
6:     Add positional encoding
7:     $\mathbf{H}' = (1 - \mathbf{m}) \odot \mathbf{H} = \mathbf{0} \in \mathbb{R}^{D \times d}$ ;           ▷ All token embedding are set zero
8:     $\mathbf{Z} = \text{Transformers}(\mathbf{H}')$
9:     Sample $\tilde{x}^{o_i}$ from $p(x^{o_i}|\mathbf{z}^{o_i})$ ;                      ▷ Sample for location $o_i$
10:    **if** $i \in \mathcal{C}$ **then**
11:        Compute $\hat{\mathbf{x}}^{o_i} = f_{o_i}(\mathbf{z}^{o_i}) \in \mathbb{R}^{\mathcal{C}_i}$
12:        Sample $\tilde{x}^{o_i} \sim \text{Multi}(\tilde{\mathbf{x}}^{o_i})$ ;              ▷ Sample discrete component
13:    **else**
14:        Sample $\tilde{x}^{o_i}$ from the diffusion sampler ;           ▷ Sample continuous component
15:    **end if**
16:    $\tilde{\mathbf{h}}^{o_i} = \mathbf{W}^{o_i} \tilde{x}^{o_i}$ ;                    ▷ Tokenization for the newly sampled column value
17:    $\mathbf{H}'_{o_i} \leftarrow \tilde{\mathbf{h}}^{o_i}$ ;                                  ▷ Update the input
18:    $m^{o_i} \leftarrow 1$ ;                                         ▷ Update the mask
19: **end for**
20: The finally sampled data is $\tilde{\mathbf{x}} = (\tilde{x}^1, \tilde{x}^2, \cdots, \tilde{x}^D)$

---

**Algorithm 5:** TABDAR: Conditional Generation (Imputation)

1: **Input:**. A generation order $\mathbf{o} = [o_1, \cdots, o_{D-M} \cdots, o_D] = \text{Shuffle}([1, 2, 3, \cdots D])$, such that $o_1, \cdots, o_{D-M}$ are the indices of $D - M$ observed columns, $o_{D-M+1}, \cdots, o_D$ are the indices of $M$ columns to impute. $x^{o_1}, \cdots x^{o_{D-M}}$ are the $D - M$ observed values.
2: **Output:** A sample $\tilde{\mathbf{x}} \in \mathbb{R}^D$, such that $\tilde{x}^{o_1} = x^{o_1}, \cdots, \tilde{x}^{o_{D-M}} = x^{o_{D-M}}$
3:
4: **for** $i \in 1, \cdots, D$ **do**
5:     **if** $i <= D - M$ **then**
6:         $\tilde{x}^{o_i} = x^{o_i}$
7:     **else**
8:         $\tilde{x}^{o_i} = 0$
9:     **end if**
10:    $\mathbf{h}^{o_i} = \mathbf{W}^{o_i} \tilde{x}^{o_i}$
11: **end for**
12: Initialize the mask vector $\mathbf{m} = [0^{D-M}, 1^M]$ ;           ▷ Tokens at missing positions are masked
13:
14: **for** $i \in D - M + 1, \cdots, D$ **do**
15:    $\mathbf{H}' = (1 - \mathbf{m}) \odot \mathbf{H}$
16:    $\mathbf{Z} = \text{Transformers}(\mathbf{H}')$
17:    Sample $\tilde{x}^{o_i}$ from $p(x^{o_i}|\mathbf{z}^{o_i})$ ;                      ▷ Sample for location $o_i$
18:    **if** $i \in \mathcal{C}$ **then**
19:        Compute $\hat{\mathbf{x}}^{o_i} = f_{o_i}(\mathbf{z}^{o_i}) \in \mathbb{R}^{\mathcal{C}_i}$
20:        Sample $\tilde{x}^{o_i} \sim \text{Multi}(\tilde{\mathbf{x}}^{o_i})$ ;              ▷ Sample discrete component
21:    **else**
22:        Sample $\tilde{x}^{o_i}$ from the diffusion sampler ;           ▷ Sample continuous component
23:    **end if**
24:    $\tilde{\mathbf{h}}^{o_i} = \mathbf{W}^{o_i} \tilde{x}^{o_i}$ ;                    ▷ Tokenization for the newly sampled column value
25:    $\mathbf{H}'_{o_i} \leftarrow \tilde{\mathbf{h}}^{o_i}$ ;                                  ▷ Update the input
26:    $m^{o_i} \leftarrow 1$ ;                                         ▷ Update the mask
27: **end for**
28: The finally sampled data is $\tilde{\mathbf{x}} = (\tilde{x}^1, \tilde{x}^2, \cdots, \tilde{x}^D)$

---

# B    DIFFUSION SDEs

This paper adopts the simplified version of the Variance-Exploding SDE in (Song et al., 2021). Song et al. (2021) has proposed the following general-form forward SDE:

$$\mathrm{d}\mathbf{x} = \boldsymbol{f}(\mathbf{x}, t)\mathrm{d}t + g(t)\,\mathrm{d}\boldsymbol{w}_t = \mathrm{d}\mathbf{x} = f(t)\,\mathbf{x}\,\mathrm{d}t + g(t)\,\mathrm{d}\boldsymbol{w}_t. \tag{9}$$

Then the conditional distribution of $\mathbf{x}_t$ given $\mathbf{x}_0$ (named as the perturbation kernel of the SDE) could be formulated as:

$$p(\mathbf{x}_t|\mathbf{x}_0) = \mathcal{N}(\mathbf{x}_t; s(t)\mathbf{x}_0, s^2(t)\sigma^2(t)\boldsymbol{I}), \tag{10}$$

where

$$s(t) = \exp\left(\int_0^t f(\xi)\mathrm{d}\xi\right), \text{ and } \sigma(t) = \sqrt{\int_0^t \frac{g^2(\xi)}{s^2(\xi)}\mathrm{d}\xi}. \tag{11}$$

Therefore, the forward diffusion process could be equivalently formulated by defining the perturbation kernels (via defining appropriate $s(t)$ and $\sigma(t)$).

Variance Exploding (VE) implements the perturbation kernel Eq. 10 by setting $s(t) = 1$, indicating that the noise is directly added to the data rather than weighted mixing. Therefore, The noise variance (the noise level) is totally decided by $\sigma(t)$. When $s(t) = 1$, the perturbation kernels become:

$$p(\mathbf{x}_t|\mathbf{x}_0) = \mathcal{N}(\mathbf{x}_t; \mathbf{0}, \sigma^2(t)\mathbf{I}) \Rightarrow \mathbf{x}_t = \mathbf{x}_0 + \sigma(t)\boldsymbol{\varepsilon}, \tag{12}$$

which aligns with the forward diffusion process in Eq. 2.

The sampling process of diffusion SDE is given by:

$$\mathrm{d}\mathbf{x} = [\boldsymbol{f}(\mathbf{x}, t) - g^2(t)\nabla_{\mathbf{x}}\log p_t(\mathbf{x})]\mathrm{d}t + g(t)\mathrm{d}\boldsymbol{w}_t. \tag{13}$$

For VE-SDE, $s(t) = 1 \Leftrightarrow \boldsymbol{f}(\mathbf{x}, t) = f(t) \cdot \mathbf{x} = \mathbf{0}$, and

$$\sigma(t) = \sqrt{\int_0^t g^2(\xi)\mathrm{d}\xi} \Rightarrow \int_0^t g^2(\xi)\mathrm{d}\xi = \sigma^2(t),$$

$$g^2(t) = \frac{\mathrm{d}\sigma^2(t)}{\mathrm{d}t} = 2\sigma(t)\dot{\sigma}(t), \tag{14}$$

$$g(t) = \sqrt{2\sigma(t)\dot{\sigma}(t)}.$$

Plugging $g(t)$ into Eq. 13, the reverse process in Eq. 3 is recovered:

$$\mathrm{d}\mathbf{x}_t = -2\sigma(t)\dot{\sigma}(t)\nabla_{\mathbf{x}_t}\log p(\mathbf{x}_t)\mathrm{d}t + \sqrt{2\sigma(t)\dot{\sigma}(t)}\mathrm{d}\boldsymbol{\omega}_t. \tag{15}$$

# C    DETAILED MODEL ARCHITECTURES

In this section, we introduce the detailed architecture of TABDAR, which consists of a tokenization layer, several transformer blocks, and two (group of) predictors for numerical and categorical features, respectively.

## C.1    TOKENIZERS AND TRANSFORMERS

**Tokenization.**    We first apply one-hot encoding to the categorical features and then project both numerical and categorical features into the embedding space. We employ column-wise tokenizers for the feature of every single column, respectively, following the setup in Zhang et al. (2024b).

- For a numerical column, we use a simple linear projection to map the scalar into a $d$-dimensional vector.
$$\mathbf{h}^i = \mathbf{W}^i x^i, \text{ where } \mathbf{W}^i \in \mathbb{R}^{1\times d} \tag{16}$$

- For a categorical column, we use a simple linear projection to map the one-hot encoding of $x^i$ into a $d$-dimensional vector.
$$\mathbf{h}^i = \mathbf{W}^i x^i, \text{ where } \mathbf{W}^i \in \mathbb{R}^{\mathcal{C}_i \times d} \tag{17}$$

**Transformer layers** After tokenization, we add column-wise positional encoding to each token embedding, and then we apply a]the zero mask to the predefined masked/target tokens. Furthermore, we append the `[pad]` token embedding at the beginning of the obtained data sequence. The proposed data will be further processed by a series of Transformer blocks.

We use ViT (Dosovitskiy, 2021) as the backbone of the Transformer layers, which consists of multiple Transformer blocks (Vaswani et al., 2017). Each Transformer block contains a multi-head self-attention mechanism and a feed-forward network. Specifically, we use a stack of six Transformer blocks with four attention heads.

**Predictors.** Given the output token embeddings from the Transformer layers, i.e., $\mathbf{z}^i$, we further use additional predictors such that it is tailored for learning the conditional distribution $p(x^i|\mathbf{z}^i)$

We apply a simple MLP predictor $f_i(\cdot)$ for each token. For each discrete column, $f_i(\cdot)$ is a 4-layer MLP with ReLU activation.

For each continuous column, the output token embedding $\mathbf{z}^i$ be further fed into the denoising neural network to predict the noise. We defer this part to Appendix C.2.

## C.2 DIFFUSION MODEL

In this section, we introduce the architecture of the diffusion model. In a nutshell, we use simple MLPs as our denoising neural network, which is similar to the design in Zhang et al. (2024b) and (Kotelnikov et al., 2023), the only difference is our denoising network takes an additional input $\mathbf{z}^i$ as the conditional information.

**Denoising neural network.** The denoising neural network takes three inputs: the noisy data $x_t^i = x^i + \sigma(t) \cdot \varepsilon$ (note that we let $\sigma(t) = t$), the current timestep $t$, and the conditional information $\mathbf{z}^i$. Following the practice in (Kotelnikov et al., 2023), the current timestep $t$ is first embedded with sinusoidal positional embedding, then further embedded with a 2-layer fully connected MLP with SiLU activation;

$$e_{pe} = \text{PE}(t), e_t = \text{MLP}_t(e_{pe})$$

where PE denotes the sinusoidal positional embedding (Vaswani et al., 2017).

Similarly, the conditional information $\mathbf{z}^i$ is embedded with a similar 2-layer MLP with SiLU activation.

$$e_z = \text{MLP}_z(\mathbf{z}^i)$$

Now we have the embeddings of current timestep and conditional information, we add them together as the final condition embedding $e^*$.

$$e^* = e_t + e_c$$

For the noisy data $x_t^i$, we first project it with a single linear layer to align the dimension with the conditional embedding, then add it with $e^*$ and feed into a 4-layer MLP with SiLU activation for the final prediction of the denoised data.

$$\boldsymbol{\epsilon}_\theta(x_t^i, t, \mathbf{z}^i) = \text{MLP}(\text{Linear}(x_t^i) + e^*)$$

Finally, we minimize the MSE loss between the output of the denoising neural network and the added noise:

$$\min \|\boldsymbol{\epsilon}_\theta(x_t^i, t, \mathbf{z}^i) - \varepsilon\|_2^2$$

# D DETAILED EXPERIMENTAL SETUPS

## D.1 HARDWARE SPECIFICATION AND ENVIRONMENT

We run our experiments on a single machine with Intel i9-14900K, Nvidia RTX 4090 GPU with 24 GB memory. The code is written in Python 3.10.14 and we use PyTorch 2.2.2 on CUDA 12.2 to train the model on the GPU.

## D.2 DATASETS

The dataset used in this paper could be automatically downloaded using the script in the provided code. We use 10 tabular datasets from Kaggle[3] or UCI Machine Learning Repository[4]: Adult[5], Default[6], Shoppers[7], Magic[8], Beijing[9], and News[10], California[11], Letter[12], Car[13], and Nursery[14], which contains varies number of numerical and categorical features. The statistics of the datasets are presented in Table 7.

Table 7: Dataset statistics.

| Dataset | # Rows | # Continuous | # Discrete | # Target | # Train | # Test | Task |
|---|---|---|---|---|---|---|---|
| **California** | 20,640 | 9 | - | 1 | 18,390 | 2,520 | Classification |
| **Letter** | 20,000 | 16 | - | 1 | 18,000 | 2,000 | Classification |
| **Car** | 1,728 | - | 7 | 1 | 1,555 | 173 | Classification |
| **Nursery** | 12,960 | - | 9 | 1 | 11,664 | 1,296 | Classification |
| **Adult** | 32,561 | 6 | 8 | 1 | 22,792 | 16,281 | Classification |
| **Default** | 30,000 | 14 | 10 | 1 | 27,000 | 3,000 | Classification |
| **Shoppers** | 12,330 | 10 | 7 | 1 | 11,098 | 1,232 | Classification |
| **Magic** | 19,021 | 10 | 1 | 1 | 17,118 | 1,903 | Classification |
| **Beijing** | 43,824 | 7 | 5 | 1 | 39,441 | 4,383 | Regression |
| **News** | 39,644 | 46 | 2 | 1 | 35,679 | 3,965 | Regression |

In Table 7, # Rows denote the number of rows (records) in the table. # Continuous and # Discrete denote the number of continuous features and discrete features, respectively. Note that there is an additional # Target column. The target columns are either continuous or discrete, depending on the task type. All datasets (except Adult) are split into training and testing sets with the ratio $9:1$ with a fixed random seed. As Adult has its official testing set, we directly use it as the testing set. For Machine Learning Efficiency (MLE) evaluation, the training set will be further split into training and validation split with the ratio $8:1$.

## D.3 TABDAR IMPLEMENTATION DETAILS

**Data Preprocessing.** We first fill the missing values with the columns's average for numerical columns. For categorical columns, missing cells are treated as an additional category. Then numerical columns are transformed to follow a normal distribution by QuantileTransformer[15] and categorical columns are encoded as integers by OrdinalEncoder[16]. Finally, we normalize the numerical features to have 0 mean and 0.5 variance, following (Karras et al., 2022).

---

[3] https://www.kaggle.com
[4] https://archive.ics.uci.edu/datasets
[5] https://archive.ics.uci.edu/dataset/2/adult
[6] https://archive.ics.uci.edu/dataset/350/default+of+credit+card+clients
[7] https://archive.ics.uci.edu/dataset/468/online+shoppers+purchasing+intention+dataset
[8] https://archive.ics.uci.edu/dataset/159/magic+gamma+telescope
[9] https://archive.ics.uci.edu/dataset/381/beijing+pm2+5+data
[10] https://archive.ics.uci.edu/dataset/332/online+news+popularity
[11] https://www.kaggle.com/datasets/camnugent/california-housing-prices
[12] https://archive.ics.uci.edu/dataset/59/letter+recognition
[13] https://archive.ics.uci.edu/dataset/19/car+evaluation
[14] https://archive.ics.uci.edu/dataset/76/nursery
[15] https://scikit-learn.org/stable/modules/generated/sklearn.preprocessing.QuantileTransformer.html
[16] https://scikit-learn.org/stable/modules/generated/sklearn.preprocessing.OrdinalEncoder.html

**Hyperparameters.** TABDAR **uses a fixed set of hyperparameters for all datasets.** Table 8 shows the hyperparameters. Our experiments show that TABDAR is robust to the choice of hyperparameters, saving the time of meticulous hyperparameter tuning for each dataset.

| Type | Parameter | Value |
|---|---|---|
| Training | optimizer | Adam |
| | initial learning rate | 1e-3 |
| | weight decay | 1e-6 |
| | LR scheduler | ReduceLROnPlateau |
| | training epochs | 5000 |
| | batch size | 4096 |
| Transformers | #Transformer blocks | 6 |
| | embedding dim | 32 |
| | #heads | 4 |

Table 8: Default hyperparameter setting of TABDAR.

### D.4 MISSING VALUE IMPUTATION

Following (Zhang et al., 2024a), we use the *Expected A Posteriori* (EAP) estimator to impute the missing values. Specifically, denote a sample $\mathbf{x}$ with missing values as $\mathbf{x} = (\mathbf{x}_{\text{obs}}, \mathbf{x}_{\text{mis}})$ where $\mathbf{x}_{\text{obs}}$ and $\mathbf{x}_{\text{mis}}$ are the observed and missing values, respectively. Our goal is to compute the expectation of the missing values conditioned on the observed values:

$$\mathbb{E}_{\mathbf{x}_{\text{mis}} \sim p(\mathbf{x}_{\text{mis}}|\mathbf{x}_{\text{obs}})}\big[\mathbf{x}_{\text{mis}}\big]$$

To estimate the expectation, we sample $k$ times from the posterior distribution $p(\mathbf{x}_{\text{mis}}|\mathbf{x}_{\text{obs}})$ (with Algorithm 5) and obtain a set of samples $\{\hat{\mathbf{x}}_{\text{mis}}^i\}_{i=1}^k$.

For numerical features, the imputation $\hat{\mathbf{x}}_{\text{mis}}$ is set to the mean of the samples:

$$\hat{\mathbf{x}}_{\text{mis}} = \frac{1}{k} \sum_{i=1}^{k} \hat{\mathbf{x}}_{\text{mis}}^i$$

For categorical features, we apply a majority vote on every column to approximate the expectation. Suppose the column has categories $\mathcal{C} = \{c_1, c_2, \cdots, c_m\}$, the imputation $\hat{\mathbf{x}}_{\text{mis}}$ is set to the most frequent category:

$$\hat{\mathbf{x}}_{\text{mis}} = \arg\max_{c_j \in \mathcal{C}} \frac{1}{k} \sum_{i=1}^{k} \delta(\hat{\mathbf{x}}_{\text{mis}}^i = c_j)$$

where $\delta(\cdot)$ is an indicator function that equals to $1$ if the input is true and $0$ otherwise. The majority vote can be understood as a mean estimator that is more robust to outliers.

Finally, we concatenate the observed values with the imputed missing values to obtain the final imputed sample:

$$\mathbf{x}_{\text{imputed}} = (\mathbf{x}_{\text{obs}}, \hat{\mathbf{x}}_{\text{mis}})$$

In our experiments, we find that $k = 10$ is a good trade-off between performance and efficiency.

### D.5 BASELINE IMPLEMENTATIONS

**Tab-MT and DP-TBART.** Tab-MT (Gulati & Roysdon, 2023) and DP-TBART (Castellon et al., 2023) are two recently proposed tabular data generation models based on MAE. To handle numerical features (with continuous distribution), Tab-MT quantizes the numerical features into 100 uniform bins, and DP-TBART quantizes the numerical features into 100 bins where each bin has the same nearest center determined by K-means. Additionally, DP-TBART employs DP-SGD (Abadi et al., 2016) to enhance the differential privacy performance. Since the focus of this paper is not on differential privacy, in our implementation, we use Adam (Kingma & Ba, 2015) optimizer.

**Other Baselines.** The implementations of SMOTE (Chawla et al., 2002), CTGAN (Xu et al., 2019), TVAE (Xu et al., 2019), GOOGLE[17] (Liu et al., 2023), GReaT (Borisov et al., 2023), CoDi (Lee et al., 2023), STaSy (Kim et al., 2023), TabDDPM (Kotelnikov et al., 2023), TabSyn (Zhang et al., 2024b) follows the codebase of Zhang et al. (2024b)[18].

## D.6 METRICS

Most of the metrics (including Marginal, Joint, $\alpha$-Precision, $\beta$-Recall, C2ST, MLE, and DCR) used in this paper directly follow the setups in Zhang et al. (2024b). Here is a reference:

- Marginal: Appendix E.3.1 in Zhang et al. (2024b).

- Joint: Appendix E.3.2 in Zhang et al. (2024b).

- $\alpha$-Precision and $\beta$-Recall: Appendix F.2 in Zhang et al. (2024b).

- C2ST: Appendix F.3 in Zhang et al. (2024b).

- MLE: Appendix E.4 in Zhang et al. (2024b).

- DCR: Appendix F.6 in Zhang et al. (2024b).

Below is a summary of how these metrics work.

### D.6.1 MARGINAL DISTRIBUTION

The **Marginal** metric evaluates if each column's marginal distribution is faithfully recovered by the synthetic data. We use Kolmogorov-Sirnov Test for continuous data and Total Variation Distance for discrete data.

**Kolmogorov-Sirnov Test (KST)** Given two (continuous) distributions $p_r(x)$ and $p_s(x)$ ($r$ denotes real and $s$ denotes synthetic), KST quantifies the distance between the two distributions using the upper bound of the discrepancy between two corresponding Cumulative Distribution Functions (CDFs):

$$\text{KST} = \sup_x |F_r(x) - F_s(x)|, \tag{18}$$

where $F_r(x)$ and $F_s(x)$ are the CDFs of $p_r(x)$ and $p_s(x)$, respectively:

$$F(x) = \int_{-\infty}^{x} p(x)\mathrm{d}x. \tag{19}$$

**Total Variation Distance (TVD)** TVD computes the frequency of each category value and expresses it as a probability. Then, the TVD score is the average difference between the probabilities of the categories:

$$\text{TVD} = \frac{1}{2} \sum_{\omega \in \Omega} |R(\omega) - S(\omega)|, \tag{20}$$

where $\omega$ describes all possible categories in a column $\Omega$. $R(\cdot)$ and $S(\cdot)$ denotes the real and synthetic frequencies of these categories.

### D.6.2 JOINT DISTRIBUTION

The **Joint** metric evaluates if the correlation of every two columns in the real data is captured by the synthetic data.

---

[17]We find the result of GOOGLE is hard to reproduce due to memory issues, so we directly use the results in Zhang et al. (2024b)

[18]https://github.com/amazon-science/tabsyn/tree/main/baselines

**Pearson Correlation Coefficient**    The Pearson correlation coefficient measures whether two continuous distributions are linearly correlated and is computed as:

$$\rho_{x,y} = \frac{\text{Cov}(x,y)}{\sigma_x \sigma_y}, \tag{21}$$

where $x$ and $y$ are two continuous columns. Cov is the covariance, and $\sigma$ is the standard deviation.

Then, the performance of correlation estimation is measured by the average differences between the real data's correlations and the synthetic data's corrections:

$$\text{Pearson Score} = \frac{1}{2}\mathbb{E}_{x,y}|\rho^R(x,y) - \rho^S(x,y)|, \tag{22}$$

where $\rho^R(x,y)$ and $\rho^S(x,y))$ denotes the Pearson correlation coefficient between column $x$ and column $y$ of the real data and synthetic data, respectively. As $\rho \in [-1,1]$, the average score is divided by 2 to ensure that it falls in the range of $[0,1]$, then the smaller the score, the better the estimation.

**Contingency similarity**    For a pair of categorical columns $A$ and $B$, the contingency similarity score computes the difference between the contingency tables using the Total Variation Distance. The process is summarized by the formula below:

$$\text{Contingency Score} = \frac{1}{2}\sum_{\alpha \in A}\sum_{\beta \in B}|R_{\alpha,\beta} - S_{\alpha,\beta}|, \tag{23}$$

where $\alpha$ and $\beta$ describe all the possible categories in column $A$ and column $B$, respectively. $R_{\alpha,\beta}$ and $S_{\alpha,\beta}$ are the joint frequency of $\alpha$ and $\beta$ in the real data and synthetic data, respectively.

### D.6.3    $\alpha$-Precision and $\beta$-Recall

$\alpha$-Precision and $\beta$-Recall are two sample-level metrics quantifying how faithful the synthetic data is proposed in Alaa et al. (2022). In general, $\alpha$-Precision evaluates the fidelity of synthetic data – whether each synthetic example comes from the real-data distribution, $\beta$-Recall evaluates the coverage of the synthetic data, e.g., whether the synthetic data can cover the entire distribution of the real data (In other words, whether a real data sample is close to the synthetic data).

### D.6.4    Classifier-Two-Sample-Test (C2ST)

C2ST studies how difficult it is to distinguish real data from synthetic data, therefore evaluating whether synthetic data can recover real data distribution. The C2ST metric used in this paper is implemented by the SDMetrics[19] package.

### D.6.5    Machine Learning Efficiency (MLE)

In MLE, each dataset is first split into the real training and testing set. The generative models are learned on the real training set. After the models are learned, a synthetic set of equivalent size is sampled.

The performance of synthetic data on MLE tasks is evaluated based on the divergence of test scores using the real and synthetic training data. Therefore, we first train the machine learning model on the real training set, split into training and validation sets with a $8:1$ ratio. The classifier/regressor is trained on the training set, and the optimal hyperparameter setting is selected according to the performance on the validation set. After the optimal hyperparameter setting is obtained, the corresponding classifier/regressor is retrained on the training set and evaluated on the real testing set. We create 20 random splits for training and validation sets, and the performance reported is the mean of the AUC/RMSE score over the 20 random trails. The performance of synthetic data is obtained in the same way.

---

[19]https://docs.sdv.dev/sdmetrics/metrics/metrics-in-beta/detection-single-table

### D.6.6 DISTANCE TO CLOSEST RECORD

We follow the 'synthetic vs. holdout' setting [20]. We initially divide the dataset into two equal parts: the first part served as the training set for training our generative model, while the second part was designated as the holdout set, which is not used for training. After completing model training, we sample a synthetic set of the same size as the training set (and the holdout set).

We then calculate the DCR scores for each sample in the synthetic set concerning both the training set and the holdout set. We further calculate the probability that a synthetic sample is closer to the training set (rather than the holdout set). When this probability is close to 50% (i.e., 0.5), it indicates that the distribution of distances between synthetic and training instances is very similar (or at least not systematically smaller) than the distribution of distances between synthetic and holdout instances, which is a positive indicator in terms of privacy risk.

## E ADDITIONAL EXPERIMENTAL RESULTS

In this section, we provide a more detailed empirical comparison between the proposed TABDAR and other baseline methods.

### E.1 DETAILED RESULTS ON THE FIDELITY METRICS

Note that in Table 1, we only present the average performance of each method on the six fidelity metrics across the ten datasets. In this section, we present a detailed performance comparison of each individual dataset:

- Marginal Distribution: Table 9
- Joint Correlation: Table 10
- $\alpha$-Precision: Table 11
- $\beta$-Recall: Table 12
- Classifier-Two-Sample-Test: Table 13
- Jensen-Shannon Divergence: Table 14

Table 9: Performance comparison on the **Marginal** Distribution Density metric. Numbers represent the error rate in %, the lower the better.

| Method | Continuous only | | Discrete only | | Heterogeous | | | | | |
|---|---|---|---|---|---|---|---|---|---|---|
| | California | Letter | Car | Nursery | Adult | Beijing | Default | Magic | News | Shoppers |
| *interpolation* | | | | | | | | | | |
| SMOTE | 0.99 | 0.97 | 1.00 | 0.57 | 1.59 | 1.78 | 1.49 | 1.07 | 5.28 | 2.48 |
| *VAE-based* | | | | | | | | | | |
| TVAE | 5.37 | 16.70 | 24.12 | 9.81 | 24.32 | 25.13 | 9.94 | 4.39 | 18.48 | 23.93 |
| *GAN-based* | | | | | | | | | | |
| CTGAN | 12.84 | 18.79 | 16.46 | 12.33 | 19.32 | 21.98 | 18.25 | 5.69 | 13.90 | 25.71 |
| *LLM-based* | | | | | | | | | | |
| GReaT | 14.93 | 4.88 | 2.22 | 5.08 | 12.12 | 8.25 | 19.94 | 16.16 | − | 14.51 |
| *Diffusion-based* | | | | | | | | | | |
| STaSy | 10.82 | 11.93 | 24.38 | 10.93 | 10.41 | 6.38 | 11.34 | 13.02 | 8.54 | 16.14 |
| CoDi | 18.98 | 22.62 | 1.53 | 0.65 | 24.84 | 12.54 | 16.54 | 11.64 | 28.13 | 36.48 |
| TabDDPM | 57.34 | 61.43 | 1.53 | 0.65 | 1.32 | 1.20 | 7.59 | 1.09 | − | 2.86 |
| TabSyn | 1.00 | 2.53 | 2.48 | 1.04 | 2.75 | 2.43 | 0.95 | 0.79 | 1.77 | 1.52 |
| *Autoregressive* | | | | | | | | | | |
| DP-TBART | 3.30 | 4.46 | 1.98 | 0.53 | 1.17 | 2.68 | 5.03 | 3.90 | 6.28 | 3.05 |
| Tab-MT | 5.87 | 3.29 | 0.96 | 0.70 | 17.20 | 25.10 | 25.17 | 21.88 | 46.54 | 2.20 |
| TABDAR | 0.99 | 1.79 | 1.31 | 0.73 | 0.59 | 0.80 | 1.74 | 0.80 | 2.03 | 1.32 |

---

[20] https://www.clearbox.ai/blog/2022-06-07-synthetic-data-for-privacy-\preservation-part-2

Table 10: Performance comparison on the **Joint** Column Correlation metric. Numbers represent the error rate in %, the low the better.

| Method | Continuous only | | Discrete only | | Heterogeous | | | | | |
|---|---|---|---|---|---|---|---|---|---|---|
| | **California** | **Letter** | **Car** | **Nursery** | **Adult** | **Beijing** | **Default** | **Magic** | **News** | **Shoppers** |
| *interpolation* | | | | | | | | | | |
| SMOTE | 2.70 | 1.19 | 3.16 | 1.21 | 3.56 | 1.53 | 6.93 | 2.84 | 2.87 | 3.53 |
| *VAE-based* | | | | | | | | | | |
| TVAE | 5.85 | 5.28 | 38.66 | 18.34 | 36.65 | 31.12 | 19.37 | 4.46 | 6.45 | 20.12 |
| *GAN-based* | | | | | | | | | | |
| CTGAN | 14.49 | 11.40 | 25.63 | 18.14 | 27.35 | 27.08 | 30.52 | 5.04 | 5.22 | 24.24 |
| *LLM-based* | | | | | | | | | | |
| GReaT | 9.66 | 3.46 | 4.72 | 8.38 | 17.59 | 59.60 | 70.02 | 59.96 | – | 45.16 |
| *Diffusion-based* | | | | | | | | | | |
| STaSy | 3.59 | 5.34 | 36.40 | 15.02 | 13.50 | 8.71 | 10.65 | 5.58 | 3.06 | 15.29 |
| CoDi | 6.89 | 5.25 | 3.52 | 1.31 | 22.72 | 6.42 | 67.88 | 6.93 | 10.81 | 20.18 |
| TabDDPM | 19.83 | 22.35 | 3.52 | 1.31 | 2.50 | 3.31 | 11.55 | 0.67 | – | 6.23 |
| TabSyn | 0.78 | 1.78 | 4.28 | 1.85 | 4.64 | 4.16 | 3.30 | 0.91 | 1.43 | 2.18 |
| *Autoregressive* | | | | | | | | | | |
| DP-TBART | 2.52 | 1.94 | 3.54 | 1.19 | 2.50 | 2.55 | 6.70 | 1.73 | 1.60 | 2.83 |
| Tab-MT | 5.87 | 3.29 | 0.96 | 0.70 | 17.20 | 25.10 | 25.17 | 21.88 | 46.54 | 2.20 |
| TABDAR | 0.61 | 1.45 | 2.99 | 1.36 | 1.36 | 2.27 | 2.83 | 1.86 | 1.50 | 1.89 |

Table 11: Performance comparison on the $\alpha$-**Precision** metric. Numbers represent $1 - \alpha$-**Precision**. The lower the better. Note that the numbers in Table 1 are in % while numbers in this table are in raw scale.

| Method | Continuous only | | Discrete only | | Heterogeous | | | | | |
|---|---|---|---|---|---|---|---|---|---|---|
| | **California** | **Letter** | **Car** | **Nursery** | **Adult** | **Beijing** | **Default** | **Magic** | **News** | **Shoppers** |
| *interpolation* | | | | | | | | | | |
| SMOTE | 0.0173 | 0.0222 | 0.0103 | 0.0040 | 0.0729 | 0.0118 | 0.0228 | 0.0186 | 0.1256 | 0.0725 |
| *VAE-based* | | | | | | | | | | |
| TVAE | 0.0191 | 0.0937 | 0.2322 | 0.0972 | 0.4124 | 0.1100 | 0.1610 | 0.0338 | 0.1530 | 0.5655 |
| *GAN-based* | | | | | | | | | | |
| CTGAN | 0.2933 | 0.0522 | 0.1023 | 0.0677 | 0.2528 | 0.0723 | 0.3595 | 0.1106 | 0.0177 | 0.1292 |
| *LLM-based* | | | | | | | | | | |
| GReaT | 0.1665 | 0.0891 | 0.0262 | 0.0836 | 0.4421 | 0.0168 | 0.1410 | 0.1454 | 1.0000 | 0.2112 |
| *Diffusion-based* | | | | | | | | | | |
| STaSy | 0.8385 | 0.0158 | 0.3347 | 0.1385 | 0.2250 | 0.0350 | 0.1320 | 0.1853 | 0.0809 | 0.1656 |
| CoDi | 0.1333 | 0.0963 | 0.0323 | 0.0037 | 0.1805 | 0.0471 | 0.1722 | 0.1434 | 0.1041 | 0.0825 |
| TabDDPM | 0.8385 | 1.0000 | 0.0323 | 0.0037 | 0.0444 | 0.0130 | 0.0924 | 0.0146 | 1.0000 | 0.0807 |
| TabSyn | 0.0062 | 0.0998 | 0.0206 | 0.0056 | 0.0239 | 0.0049 | 0.0123 | 0.0053 | 0.0457 | 0.0279 |
| *Autoregressive* | | | | | | | | | | |
| DP-TBART | 0.0256 | 0.0427 | 0.0093 | 0.0054 | 0.0054 | 0.0049 | 0.0727 | 0.0160 | 0.0149 | 0.0145 |
| Tab-MT | 0.0239 | 0.0415 | 0.0140 | 0.0146 | 0.1776 | 0.4839 | 0.0581 | 0.5507 | 0.9369 | 0.0158 |
| TABDAR | 0.0080 | 0.0383 | 0.0190 | 0.0052 | 0.0029 | 0.0070 | 0.0133 | 0.0147 | 0.0218 | 0.0025 |

## E.2 TRAINING / SAMPLING TIME

In Table 15, we compare the training and sampling time of TABDAR with other methods on the Adult dataset.

## E.3 ADDITIONAL VISUALIZATIONS

We present the 2D visualizations (Figure 7 and Figure 8) of the synthetic data generated by all baseline methods in Figure 10 and Figure 11, respectively. We also present the heat maps of all methods on the four datasets (Figure 9) in Figure 12, Figure 13, Figure 14, and Figure 15, respectively.

Table 12: Performance comparison on the $\beta$-**Recall** metric. Numbers represent $1 - \beta$-**Recall**. The lower the better. Note that the numbers in Table 1 are in % while numbers in this table are in raw scale.

| Method | Continuous only | | Discrete only | | Heterogeous | | | | | |
|---|---|---|---|---|---|---|---|---|---|---|
| | California | Letter | Car | Nursery | Adult | Beijing | Default | Magic | News | Shoppers |
| *Interpolation* | | | | | | | | | | |
| SMOTE | 0.2157 | 0.1338 | 0.0045 | 0.0012 | 0.2325 | 0.2080 | 0.2390 | 0.1934 | 0.2011 | 0.2386 |
| *VAE-based* | | | | | | | | | | |
| TVAE | 0.6512 | 0.8324 | 0.4369 | 0.2625 | 0.8969 | 0.9392 | 0.7078 | 0.6243 | 0.7413 | 0.7672 |
| *GAN-based* | | | | | | | | | | |
| CTGAN | 0.8412 | 0.9903 | 0.9879 | 0.0530 | 0.8246 | 0.6230 | 0.8889 | 0.8472 | 0.7732 | 0.7441 |
| *LLM-based* | | | | | | | | | | |
| GReaT | 0.5515 | 0.6643 | 0.0034 | 0.0024 | 0.5088 | 0.5666 | 0.5796 | 0.6509 | 1.0000 | 0.5510 |
| *Diffusion-based* | | | | | | | | | | |
| STaSy | 0.9288 | 0.7332 | 0.1075 | 0.0029 | 0.6812 | 0.5061 | 0.6421 | 0.5686 | 0.6033 | 0.7174 |
| CoDi | 0.5998 | 0.4551 | 0.0040 | 0.0009 | 0.9032 | 0.4472 | 0.7811 | 0.5139 | 0.6505 | 0.8187 |
| TabDDPM | 0.9288 | 1.0000 | 0.0040 | 0.0009 | 0.5152 | 0.4335 | 0.6150 | 0.5206 | 1.0000 | 0.4492 |
| TabSyn | 0.5706 | 0.7486 | 0.0031 | 0.0007 | 0.5484 | 0.4847 | 0.5365 | 0.5146 | 0.5602 | 0.4399 |
| *Autoregressive* | | | | | | | | | | |
| DP-TBART | 0.6138 | 0.8859 | 0.0038 | 0.0010 | 0.5033 | 0.4536 | 0.5562 | 0.6044 | 0.6532 | 0.5285 |
| Tab-MT | 0.6272 | 0.8385 | 0.0067 | 0.0008 | 0.9681 | 0.4930 | 0.9179 | 0.9844 | 1.0000 | 0.5105 |
| TABDAR | 0.5038 | 0.4562 | 0.0040 | 0.0008 | 0.4928 | 0.4249 | 0.5162 | 0.3887 | 0.5277 | 0.4078 |

Table 13: Performance comparison on the **C2ST** metric. Numbers represent $100 \times (1 - \textbf{C2ST})$ (i.e. in base of $10^{-2}$). The lower the better.

| Method | Continuous only | | Discrete only | | Heterogeous | | | | | |
|---|---|---|---|---|---|---|---|---|---|---|
| | California | Letter | Car | Nursery | Adult | Beijing | Default | Magic | News | Shoppers |
| *interpolation* | | | | | | | | | | |
| SMOTE | 0.61 | 0.00 | 0.00 | 0.00 | 3.05 | 0.44 | 7.69 | 1.93 | 6.49 | 9.80 |
| *VAE-based* | | | | | | | | | | |
| TVAE | 12.48 | 22.27 | 70.30 | 52.73 | 72.39 | 45.53 | 41.65 | 12.07 | 60.27 | 70.04 |
| *GAN-based* | | | | | | | | | | |
| CTGAN | 50.11 | 82.40 | 59.59 | 48.63 | 36.79 | 56.82 | 64.60 | 14.15 | 27.64 | 48.86 |
| *LLM-based* | | | | | | | | | | |
| GReaT | 28.38 | 16.14 | 6.05 | 18.41 | 46.24 | 31.07 | 52.90 | 56.74 | – | 57.15 |
| *Diffusion-based* | | | | | | | | | | |
| STaSy | 54.61 | 47.75 | 78.52 | 40.43 | 54.02 | 23.48 | 49.29 | 53.97 | 50.21 | 62.20 |
| CoDi | 47.48 | 42.85 | 0.00 | 0.22 | 80.02 | 15.70 | 52.37 | 27.70 | 91.62 | 81.84 |
| TabDDPM | 88.01 | 96.86 | 0.00 | 0.22 | 3.95 | 3.29 | 11.75 | 0.95 | – | 16.37 |
| TabSyn | 0.71 | 3.18 | 2.90 | 2.88 | 8.05 | 3.60 | 1.33 | 0.08 | 1.77 | 3.05 |
| *Autoregressive* | | | | | | | | | | |
| DP-TBART | 3.89 | 11.07 | 0.07 | 0.00 | 0.81 | 3.63 | 7.41 | 5.01 | 12.96 | 8.73 |
| Tab-MT | 8.71 | 9.34 | 0.37 | 0.00 | 99.86 | 99.97 | 91.62 | 74.47 | 100.00 | 1.47 |
| TABDAR | 1.27 | 0.55 | 0.00 | 0.74 | 0.47 | 1.55 | 3.43 | 0.1100 | 3.12 | 3.74 |

Table 14: Performance comparison on the Jensen-Shannon Divergence (**JSD**) metric. The lower the better. Note that the numbers in Table 1 are in % while numbers in this table are in raw scale.

| Method | Continuous only | | Discrete only | | Heterogeous | | | | | |
|---|---|---|---|---|---|---|---|---|---|---|
| | California | Letter | Car | Nursery | Adult | Beijing | Default | Magic | News | Shoppers |
| *interpolation* | | | | | | | | | | |
| SMOTE | 0.0006 | 0.0006 | 0.0013 | 0.0008 | 0.0008 | 0.0003 | 0.0006 | 0.0008 | 0.0040 | 0.0015 |
| *VAE-based* | | | | | | | | | | |
| TVAE | 0.0041 | 0.0072 | 0.0323 | 0.0132 | 0.0078 | 0.0077 | 0.0052 | 0.0036 | 0.0109 | 0.0107 |
| *GAN-based* | | | | | | | | | | |
| CTGAN | 0.0182 | 0.0135 | 0.0220 | 0.0165 | 0.0038 | 0.0033 | 0.0114 | 0.0056 | 0.0078 | 0.0066 |
| *LLM-based* | | | | | | | | | | |
| GReaT | 0.0111 | 0.0022 | 0.0030 | 0.0068 | 0.0182 | 0.0023 | 0.0076 | 0.0107 | – | 0.0056 |
| *Diffusion-based* | | | | | | | | | | |
| STaSy | 0.0380 | 0.0057 | 0.0326 | 0.0146 | 0.0041 | 0.0030 | 0.0055 | 0.0107 | 0.0070 | 0.0086 |
| CoDi | 0.0152 | 0.0085 | 0.0020 | 0.0009 | 0.0073 | 0.0017 | 0.0067 | 0.0142 | 0.0092 | 0.0103 |
| TabDDPM | 0.0380 | 0.0382 | 0.0020 | 0.0009 | 0.0004 | 0.0092 | 0.0008 | 0.0013 | – | 0.0019 |
| TabSyn | 0.0006 | 0.0017 | 0.0033 | 0.0014 | 0.0004 | 0.0012 | 0.0003 | 0.0007 | 0.0016 | 0.0007 |
| *Autoregressive* | | | | | | | | | | |
| DP-TBART | 0.0023 | 0.0010 | 0.0023 | 0.0008 | 0.0004 | 0.0006 | 0.0025 | 0.0018 | 0.0036 | 0.0008 |
| Tab-MT | 0.0038 | 0.0019 | 0.0013 | 0.0009 | 0.0026 | 0.0024 | 0.0034 | 0.0211 | 0.0221 | 0.0007 |
| TABDAR | 0.0004 | 0.0010 | 0.0018 | 0.0010 | 0.0003 | 0.0004 | 0.0007 | 0.0011 | 0.0015 | 0.0008 |

Table 15: Comparison of training and sampling time of different methods, on Adult dataset.

| Method | Training | Sampling |
|---|---|---|
| CTGAN | 1029.8s | 0.8621s |
| TVAE | 352.6s | 0.5118s |
| GReaT | 2h 27min | 2min 19s |
| STaSy | 2283s | 8.941s |
| CoDi | 2h 56min | 4.616s |
| TabDDPM | 1031s | 28.92s |
| TabSyn | 2422s | 11.84s |
| DP-TBART | 2355.8s | 2.11s |
| Tab-MT | 2352.23s | 2.18s |
| TABDAR | 2701.5s | 21.66s |

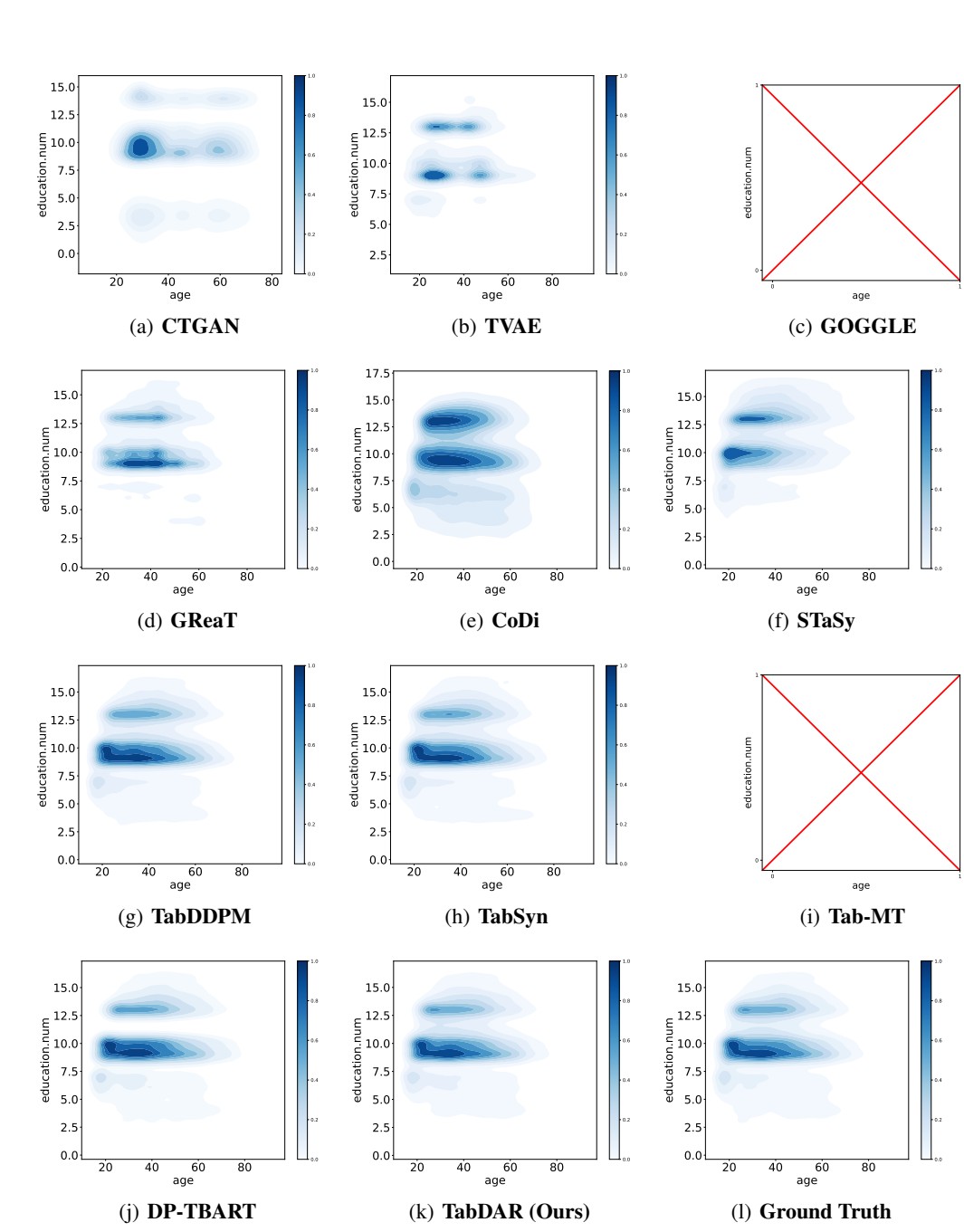

Figure 10: Kernel density estimation (KDE) plot of the 2D joint density of 'education.num' and 'age' features in the Adult dataset. The results from GOGGLE and Tab-MT are not plotted since they either fail to generate or generate singleton synthetic data on one feature (e.g. always generate 'education.num' equals one).

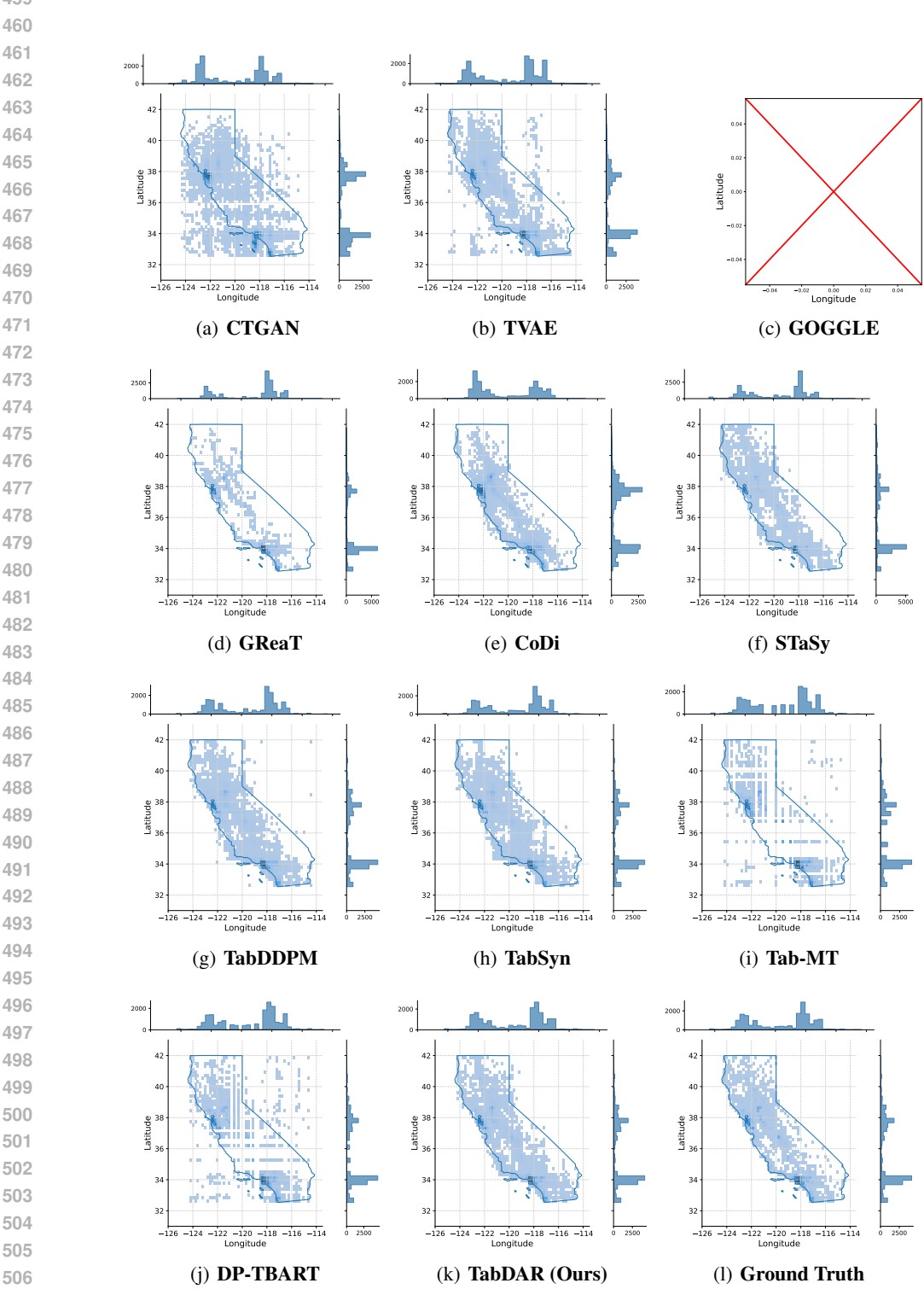

Figure 11: Scatter plots of the 2D joint density of the Longitude and Latitude features in the California Housing dataset. Blue lines represent the geographical border of California.

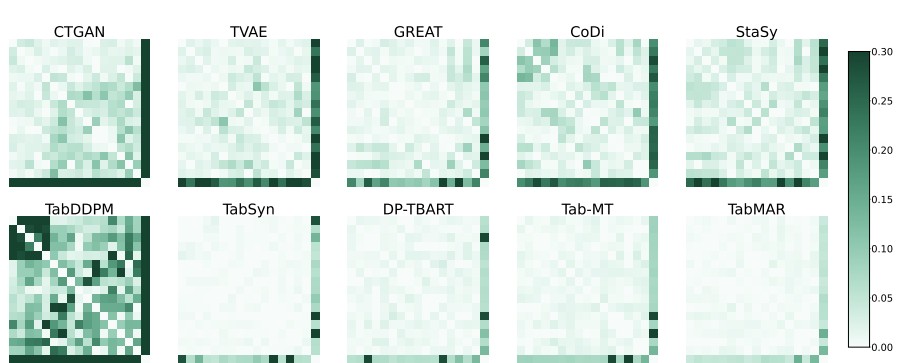

Figure 12: Heat map of synthetic data of Letter dataset.

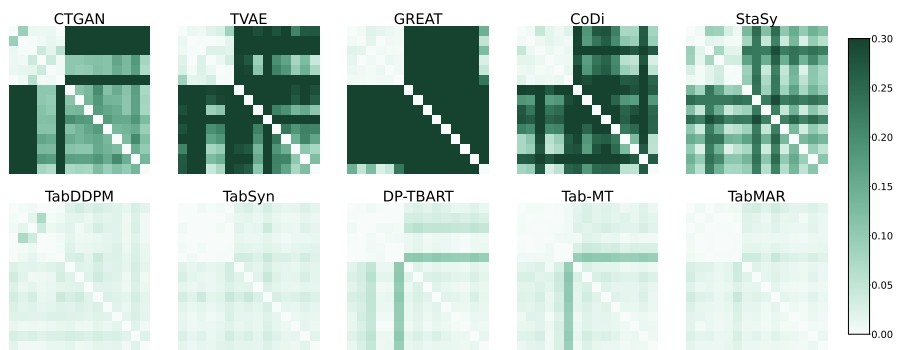

Figure 13: Heat map of synthetic data of Adult dataset.

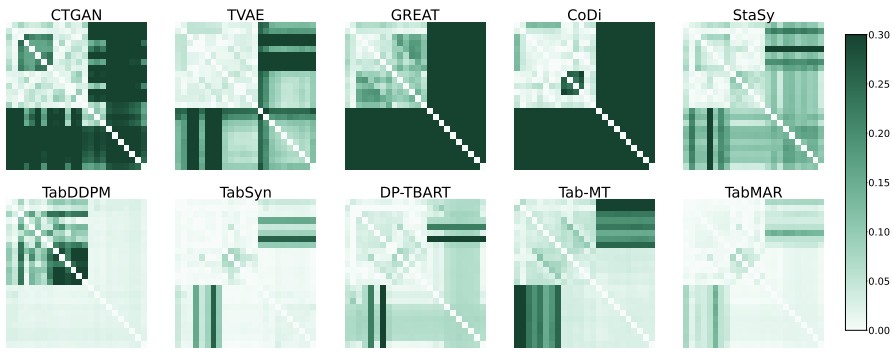

Figure 14: Heat map of synthetic data of Default dataset.

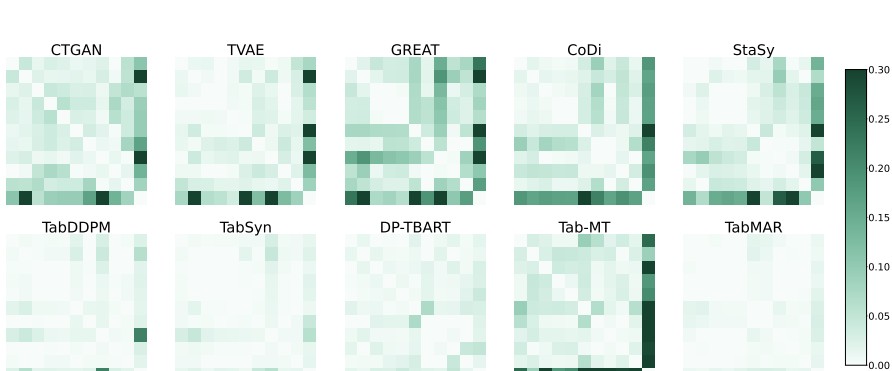

Figure 15: Heat map of synthetic data of Magic dataset.

