# OpenReview forum: "Diffusion-Nested Auto-Regressive Synthesis of Heterogeneous Tabular Data"
_ICLR.cc/2025/Conference — Submitted to ICLR 2025_

### Official Review · Reviewer_Dbtr · 2024-11-02

**Soundness:** 3
**Presentation:** 3
**Contribution:** 3
**Rating:** 5
**Confidence:** 3

**Summary:**

This paper suggests a new tabular data synthesizer by combining bi-directional transformer and diffusion model.

**Strengths:**

Strong experimental results.

**Weaknesses:**

Please see the questions below.

**Questions:**

A.	I think the data generation time will take long because of the autoregressive nature of the model. I think it would be great if authors can compare the generation times with other baseline methods.

B.	In line 85, the authors claim they use two types of generative models: one is diffusion, and another one is categorical prediction. But categorical prediction is not a generative model.

C.	Authors should mention they use one-hot-encoding for preprocessing categorical data. I noticed they mentioned it in Appendix, but they should be mentioned in main paper to avoid confusion.

D.	In Figure 3, the role of diffusion model is not emphasized.

E.	Are multiple diffusion models used for each of the continuous variable? If it is true, for each

F.	In Tables 5 & 6, are the values in the tables averaged results over the two datasets?

G.	Is it safe to say the model is robust to the hyperparameters? The results are based only on two datasets. I think authors need to work on more datasets for validating this claim.

H.	Are the P(P(x|z))s in Eq. (6) and (7) typos?

**Details Of Ethics Concerns:**

I don't see ethical concerns will arise from this paper.

---

> ### Author Response · Authors · 2024-12-01
> **Response to Reviewer Dbtr (1 of 2)**
>
> We greatly appreciate the reviewer for taking the time to provide such constructive comments and suggestions. The following are detailed responses to your questions.
>
>
> ### Generation time comparison
>
> > I think the data generation time will take a long because of the autoregressive nature of the model. I think it would be great if authors can compare the generation times with other baseline methods.
>
> Thank you for your question. When using denoising neural networks of the same scale, TabDAR's sampling rate does indeed take longer time due to its autoregressive sampling strategy, compared to conventional diffusion models (such as TabDDPM and TabSyn) that sample all columns simultaneously. However, since our TabDAR's diffusion model is applied to each column individually, only learning the conditional distribution of this single column, its denoising neural network can be much smaller (more lightweight) compared with those used in TabDDPM and TabSyn.
>
> Specifically, our TabDAR uses a denoising neural network that is a 3-layer MLP with hidden dimension = 512. In contrast, TabSyn's paper states that it uses a denoising neural network that is a 5-layer MLP with hidden dimension = 2048. In this case, our TabDAR's actual sampling speed is basically similar to TabSyn's.
>
> In the following table, we compare the time required by several methods to sample synthetic data of the same size as the original dataset on the Adult dataset. (Note that the sampling time for different datasets scales approximately linearly with the size of the dataset (number of rows and columns)).
>
> | Methods | CTGAN  | TVAE | GOGGLE | GReaT  | STaSy |  CoDi | TabDDPM | TabSyn | TabDAR (ours)
> | ------ | --------- | ------ | --------- | ------  |  ------ | --------- | ------  |  ------ | --------- |
> | Sampling time  | 0.87s | 0.59s |  5.8s   | 158s  | 9.1s | 4.7s | 26.8s | 2.1s | 2.6s|
>
> The data in the table shows that TabDAR's sampling rate is only slightly slower than TabSyn, while being much faster than other Diffusion models such as CoDi, STaSy, and TabDDPM. In terms of empirical performance, TabDAR significantly outperforms TabSyn.
>
>
> ### Categorical prediction is not a generative model
> > In line 85, the authors claim they use two types of generative models: one is diffusion, and another one is categorical prediction. However, categorical prediction is not a generative model.
>
> Thank you for your question. We humbly point out that through a categorical distribution (a 1*C vector where each entry represents the probability of belonging to each category), we clearly capture the conditional probability of a discrete column value. At the same time, we can also easily sample based on this probability distribution vector. This is the key reason why we call it a generative model.
>
> The above describes the one-dimensional case. If the data is higher-dimensional, such as text data, we can naturally combine categorical distribution with autoregressive models to capture the distribution of high-dimensional data.
>
> ### Mention one-hot encoding for preprocessing categorical data
>
> > Authors should mention they use one-hot-encoding for preprocessing categorical data. I noticed they mentioned it in Appendix, but they should be mentioned in main paper to avoid confusion.
>
> Thank you for your suggestion. In the revised paper's Tokenization layer section, we will add an explanation that our linear transformation matrix is applied to one-hot encoding for discrete variables.
>
> #### In Figure 3, the role of the diffusion model is not emphasized.
>
> Thank you for your suggestion. In the revised paper we have emphasized that a diffusion model is nested at the output of a cotinuous column in order to model its conditional distribution.
>
> ### Are multiple diffusion models used for each of the continuous variables? If it is true, for each
>
> Thank you for noticing this issue. We use one unique diffusion model for all continuous columns. We apologize for not explicitly stating this in the main text. In Equation (7), we implicitly indicated this by using the same denoising neural network notation $\varepsilon_{\theta}$ for all column indices $i$. We will clearly state this in the revised paper.
>
> ### In Tables 5 & 6, are the values in the tables averaged results over the two datasets? Model's robustness to hyperparameters.
>
> Yes. The values in Table 5 and Table 6 are the averaged performance on Adult and Beijing datasets.

---

> ### Author Response · Authors · 2024-12-01
> **Response to Reviewer Dbtr (2 of 2)**
>
> ### Model's robustness to hyperparameters
> > Is it safe to say the model is robust to the hyperparameters? The results are based only on two datasets. I think the authors need to work on more datasets to validate this claim.
>
> Thank you for your question. We have conducted additional experiments, studying the hyperparameter sensitivity of TabDAR on all datasets. The following table shows the average performance across all datasets.
>
> | Depth | Margin  | Joint | $\alpha$-Precision | $\beta$-Recall  |
> | ------ | --------- | ------ | --------- | ------  |
> | 2   | 1.92% | 2.64% |  2.02%   | 42.5\%  |
> | 4   | 1.68% | 2.16% |  1.51%   | 38.9\%  |
> | 6   | **1.21%** | **1.80%** |  **1.33%**   | **37.2\%**  |
> | 8   | 1.35% | 1.91% |  1.56%   | 39.7\%  |
>
> | Dim | Margin  | Joint | $\alpha$-Precision | $\beta$-Recall  |
> | ------ | --------- | ------ | --------- | ------  |
> | 8   | 2.03% | 2.94% |  2.24%   | 43.2\%  |
> | 16   | 1.64% | 2.26% |  1.74%   | 39.9\%  |
> | 32  | **1.21%** | **1.80%** |  **1.33%**  | **37.2\%**  |
> | 64   | 1.49% | 2.09% |  1.82%   | 37.8\%  |
>
> The experimental results show that the average performance of our TabDAR across all datasets is minimally affected by the transformer's depth and embedding dimension. Therefore, TabDAR demonstrates good robustness.
>
> ### Are the P(P(x|z))s in Eq. (6) and (7) typos?
>
> Thank you for pointing out this typo. It should be $P(x|z)$.

---

### Official Review · Reviewer_au72 · 2024-11-03

**Soundness:** 2
**Presentation:** 2
**Contribution:** 2
**Rating:** 3
**Confidence:** 3

**Summary:**

This work proposes to combine autoregressive sequence modeling for discrete (token) generation and diffusion for continuous signal.

**Strengths:**

The authors present state-of-the-art empirical results (by some margin). The proposed method is based on interesting observations and to the best of my knowledge, the way they combine multi-modal signals is novel.

**Weaknesses:**

- The authors claim that columns are permutation invariant, which I tend to agree with. However, combining this observation with autoregressive generation is impossible. From the manuscript, I couldn't figure out if the method combines lower triangular mask and permutations of the tokens. Can the authors clarify?
- Superscript and subscripts - it seems like the authors used mixed notations in different sections (I think?). E.g., in equation (1), items in a sequence are subscript and then $<i$ is in superscript; later, the authors use the subscript for diffusion time and superscript for items (section 3.1, equation 2, equation 3, equation 5, and more). It feels like a mess. Please let me know if I didn't understand correctly.
- The authors claim that it is straightforward to do conditional sampling using their method, but I find it hard to understand how it is actually done. I.e., condition on continuous representation in discrete phase and vice versa.
- Learning a permutation invariant representation is well known in transformers, and to the best of my knowledge, the classical way to do it is by removing the positional embeddings. Can the authors explain why they chose a different path (which seems less effective)?
- Equation 6 - is it correct?
- Results - I am not familiar with the benchmarks used, but I am not sure what I can learn about the proposed method compared to prior works when the improvements are so small (as it seems that the tasks are roughly solved).

**Questions:**

See above.

---

> ### Author Response · Authors · 2024-12-01
> **Response to Reviewer Au72 (1 of 2)**
>
> We greatly appreciate the reviewer for taking the time to provide such constructive comments and suggestions. The following are detailed responses to your questions.
>
> ### Column permutation invariance.
>
> > The authors claim that columns are permutation invariant, which I tend to agree with. However, combining this observation with autoregressive generation is impossible. From the manuscript, I couldn't figure out if the method combines lower triangular mask and permutations of the tokens. Can the authors clarify?
>
> Thanks for your question.
>
> During the training phase, TabDAR uses masked token distribution modeling as its training method, which is similar to masked language modeling in BERT, aiming to learn the target column's conditional distribution, conditioned on any observed columns.
>
> In the generation phase, TabDAR samples specified columns one by one according to a pre-specified generation order, then fills the sampled values into the input. Through this process of unfolding masked columns one by one, it achieves autoregressive generation. This is essentially the same as the autoregressive generation based on lower triangular masks that you mentioned. Let us explain this with an example.
>
> Suppose we have three columns, from left to right they are col1 = 'age', col2 = 'gender', col3 = 'salary'. If we want to generate data autoregressively in the form of 'age' -> 'salary' -> 'gender', we can
>
> - Route1: First shuffle these three columns, so that col1 = 'age', col2 = 'salary', col3 = 'gender'. Then add a lower triangular mask, so that the model first generates the value for 'age', then adds the generated age value to the model's input, then calls the model to generate the value for the 'salary' column, and so on.
>
> - First, generate a mask vector [0,0,0], indicating that initially all column values are unknown. Then given the generation order 1 -> 3 -> 2, sample the value for col1 based on the fully masked input. Next, fill in col1's value and set the mask vector to [1,0,0], use this as input to sample the value for col3. Then fill in col3's value and set the mask vector to [1,0,1], and finally use this as input to sample the value for the second column.
>
> Therefore, these two sampling methods achieve the same effect, but in our implementation, we no longer need to shuffle columns - we only need to operate on the mask vector to obtain the desired generated data.
>
>
> ### Superscript and subscripts
> > Superscript and subscripts - it seems like the authors used mixed notations in different sections (I think?). E.g., in equation (1), items in a sequence are subscript and then is in superscript; later, the authors use the subscript for diffusion time and superscript for items (section 3.1, equation 2, equation 3, equation 5, and more). It feels like a mess. Please let me know if I didn't understand correctly.
>
> Thank you for your criticism. We have corrected all notations throughout the paper. Now, all superscripts indicate column indices, while all subscripts indicate diffusion timesteps. Thank you again for your correction.

---

> ### Author Response · Authors · 2024-12-01
> **Response to Reviewer Au72 (2 of 2)**
>
> ### Learning permutation invariant representations by removing positional embeddings.
>
> > Learning a permutation invariant representation is well known in transformers, and to the best of my knowledge, the classical way to do it is by removing the positional embeddings. Can the authors explain why they chose a different path (which seems less effective)?
>
> Thank you for your question. We would like to address your concerns by clarifying the following two points.
>
> 1. Achieving permutation invariance by removing positional embedding is indeed common in representation learning. However, we would like to remind the reviewer that the problem studied in this paper is not representation learning (not learning the representation of a row), but rather learning a column's conditional distribution, given any observed column(s). Therefore, it is necessary to remove information from unobserved columns through the addition of masks.
>
> 2. In TabDAR, we did remove traditional position embeddings based on column order. Instead, we add column-specific token embeddings for each column. This is because although the order of columns is not important, different columns still have distinctly different underlying meanings. Therefore, adding column-specific token embeddings enables the model to distinguish between different columns.
>
>
>
>
>
>
> ### Equation 6 - is it correct?
>
> Thank you for pointing out this typo. It should be $P(x|z)$.
>
> ### The importance of the empirical results.
>
> > Results - I am not familiar with the benchmarks used, but I am not sure what I can learn about the proposed method compared to prior works when the improvements are so small (as it seems that the tasks are roughly solved).
>
> Thank you for your question. If you only look at the results in Table 2, you might think this task is roughly solved. This is because machine learning efficiency (which is established by previous literature) is itself a relatively easy downstream task, resulting in very small performance differences between different models, making them difficult to distinguish. However, on other metrics, such as the statistical fidelity metrics in Table 1, our TabDAR shows quite significant performance improvements. Additionally, since our model models the conditional distribution rather than the joint distribution, it can be conveniently used for missing data imputation tasks (see Figure 6). These are all things that TabSyn, the second-best baseline method, cannot achieve.

---

### Official Review · Reviewer_HuEq · 2024-11-03

**Soundness:** 3
**Presentation:** 3
**Contribution:** 2
**Rating:** 5
**Confidence:** 4

**Summary:**

The authors propose a Diffusion-nested Autoregressive model (TABDAR) that can handle both discrete and continuous tabular data. Continuous features are modeled by nested diffusion models, and based on the assumption that table columns are permutation invariant, bi-directional attention is utilized to support generation of the columns in any order. TABDAR is compared to several recent methods for hetrogeneous table generation, and is shown to perform comparably or better in terms of statistical fidelity (Table 1), generation efficiency for machine learning tasks (Table 2), and Distance to Closest Record (DCR) scores (Table 3, Figure 5). Ablations suggest that both the use of diffusion losses and random sampling of order improve performance (Table 4), and that the results are quite stable wrt depth and embedding dimension (Table 5 and 6). Imputation results are also competitive (Figure 6).

**Strengths:**

- Good results, and extensive experimental results.

**Weaknesses:**

- The TabSyn methods results stated in the paper are different (worse) that those reported in the TabSyn paper. These differences need to be fully explained and accounted for before the paper can be considered for publication.
- In addition, since TabSyn and TABDAR perform similarly and address the same problem, the advantages and differences between the two approaches should be discussed.
- As a method that simply integrates diffusion modeling into bidirectional transformers to handle continuous tabular entries, the novelty of the paper seems on the lower side.
- The use of bidirectional attention is based on the assumption that columns are permutation invariant, while in practice, one expects that columns will sometimes have dependencies, and these would always be based on reading order (e.g. for English, left to right). Also, bi-directional attention is generally not equivalent to arbitrarily ordered causal attention, as suggested in Figure 2, due to the use of position embeddings. Based on these observations, It's not clear that bi-directional attention necessarily the best approach.

**Questions:**

See previous section.

---

> ### Author Response · Authors · 2024-12-01
> **Response to Reviewer HuEq**
>
> We greatly appreciate the reviewer for taking the time to provide such constructive comments and suggestions. The following are detailed responses to your questions.
>
> ### Differences of the empirical performance
> > The TabSyn methods results stated in the paper are different (worse) that those reported in the TabSyn paper. These differences need to be fully explained and accounted for before the paper can be considered for publication.
>
> All our experimental results were obtained by re-running all methods on all datasets, and all results can be reproduced using the anonymous code repository mentioned in the paper. As shown in the experimental settings section, we considered more baseline methods compared to the TabSyn paper, such as TabMT and DP-TBART, and also included more datasets (e.g., California, Letter, Car, Nursery). Therefore, we had to reuse and re-run the code for all baselines.
>
> ### Differences between TabDAR and TabSyn
>
> > In addition, since TabSyn and TABDAR perform similarly and address the same problem, the advantages and differences between the two approaches should be discussed.
>
> Thank you for your questions and suggestions. There are several fundamental differences between TabSyn and the proposed TabDAR:
>
> - TabSyn consists of two submodules and is not an end-to-end model. It requires training a VAE model and then training a Diffusion model based on the VAE's latent space, while our TabDAR is an end-to-end generative model.
> - TabSyn, both in its VAE part and Diffusion part, essentially models the joint distribution of all columns in the tabular data. In contrast, the proposed TabDAR uses a fundamentally different autoregressive framework, which learns the conditional distribution of individual columns during training and generates data column by column in a given order during generation.
>
> We will explicitly state these differences in the related works section to highlight the uniqueness of our method.
>
>
> ### Novelty of the proposed method.
> > As a method that simply integrates diffusion modeling into bidirectional transformers to handle continuous tabular entries, the novelty of the paper seems on the lower side.
>
> Thank you for your question. The innovation of our method is not "combining Diffusion models with Bi-directional Transformers as a neural network architecture," but rather "embedding Diffusion models into an autoregressive generation framework." This represents a fusion of two types of generative models, where Bi-directional Transformers are simply a means to implement autoregression (as arbitrary generation orders can be achieved through masking).
>
> The way we integrate the Diffusion model into Autoregressive generation is that for continuous columns, we model their conditional distribution using a diffusion model, rather than using categorical distribution for non-continuous variables, or direct regression (since regression learns deterministic predicted values rather than a probability distribution).
>
> ### Rationale of the bidirectional attention.
> > The use of bidirectional attention is based on the assumption that columns are permutation invariant, while in practice, one expects that columns will sometimes have dependencies, and these would always be based on reading order (e.g. for English, left to right). Also, bi-directional attention is generally not equivalent to arbitrarily ordered causal attention, as suggested in Figure 2, due to the use of position embeddings. Based on these observations, It's not clear that bi-directional attention necessarily the best approach.
>
> Thank you for the reviewer's question.
>
>
> #### Dependency of columns.
> We agree with the reviewer's point that for tabular data, there are indeed strong dependencies between different columns and even causal relationships. When these causal relationships are known, we can certainly construct the most appropriate generation order based on them, and our model can focus on this fixed order during training and testing. We also believe this would help our model learn the data distribution better. However, the challenge lies in the fact that the causal relationships are unknown for most datasets. Only with very strong domain expertise or the assistance of causal discovery models can we obtain relatively accurate causal relationships. And this is already out of the scope of this paper.
>
> Therefore, when the dependencies between columns in the data itself, let alone the causal order, are unknown, we believe that considering column permutation invariance is a relatively reasonable choice.
>
> #### Bi-directional attention is generally not equivalent to arbitrarily ordered causal attention.
>
> In our setup, we add column-specific position embeddings for each column, rather than traditional position embeddings based on column order. This means that the positional embedding for each column's data won't change due to column shuffling. Therefore, the masked bi-directional attention, in our case, is equivalent to causal attention.

---

### Official Review · Reviewer_QPGs · 2024-11-04

**Soundness:** 3
**Presentation:** 2
**Contribution:** 3
**Rating:** 6
**Confidence:** 4

**Summary:**

The authors develop a generative modeling method for tabular data.  Their method learns a conditional generative model that takes masked input and predicts the missing columns.  At inference time, they use this model autoregressively to generate data one column at a time -- for both unconditional data generation and missing values imputation. The model comprises of a linear embedder, a transformer encoder, and either a diffusion decoder or categorical classifier -- depending on whether the particular column is continuous or discrete in nature.  The authors perform experiments on various tabular datasets and use various metrics to compare the samples generated by their method against those generated by other methods in the literature.  They also perform ablation studies to understand the relative importance of various components of their approach.

**Strengths:**

- This paper tackles an interesting and important problem -- how to learn the generative distribution of tabular data.  Tabular data is indeed challenging because of the potentially unstructured and diverse nature of the columns at hand.  As the authors mention, there has been a lot of recent interest in developing deep generative models for this domain.
- The authors' proposed solution is an interesting one.  They provide comparisons in the experiments against several other methods in the literature and generally show superior performance on the statistical fidelity metrics (Table 1).
- I appreciate the extensive set of experiments, such as machine learning efficiency, the various visualizations, and the ablation studies. These help give a holistic view of the relative performance of their method compared to the literature.
- The code is available and generally looks well-organized and a useful resource for the community.

**Weaknesses:**

- Nitpick: The authors sometimes use overly promotional language, e.g. line 71 — “through two ingenious design features”, line 84 —  “TABDAR offers several unparalleled advantages” — I recommend against the use of overly promotional words such as “ingenious”, “unparalleled advantages” in the paper.
 - Nitpick: Eq. (5) — Define $x^{<i}$.  For a reader familiar with autoregressive modeling notation, it is obvious what this means, but for others it may not be.  E.g. it would be good to clarify what this means for $i = 1$.
- One point that was confusing to me at first was that the model is not trained as an autoregressive model -- it is trained as a masked language model.  If I understand Eq. (8) / Algorithm 3 correctly, you are only masking out a particular set of columns at each training step.
 Thus, you are not really learning an autoregressive distribution, this is more like a conditional distribution like masked language modeling in which you predict p(masked | unmasked).  I understand that you are using the learned conditional distribution in an autoregressive way at test time / generation time -- it would be good to clarify this point in the text.
- There are also some questions about the method I have, listed below.  I am happy to give a positive score for this paper, but it is contingent on the questions below being addressed.

**Questions:**

- For continuous columns, why use a diffusion loss?  You could also use z to parameterize any simple continuous distribution, e.g. a Gaussian.  Given that the diffusion process can be computationally expensive, do you think that it is a necessary component of the method?  I understand that you do an ablation in which you compare against discretizing the continuous variable, but this also seems like an unnatural approach compared to simply passing it through a continuous distribution.
- In your embedding/masking, how do you distinguish between a continuous value being equal to 0 and you masking out that value?  To me, it seems that both would lead to the same result after embedding the data.  Is there a particular reason you chose to use zero for missing values instead of an embedding for a [mask] token, as typical in masked language models?
- Details on the diffusion aspect are generally missing.  Is the denoising network the same for different continuous columns or is the denoising network column-specific?  How many steps of diffusion are generally used?
- Table 2 — since some of these values are quite close, could you provide some standard deviations?  Also, if there are ties with your method in the table, you should probably also bold these for consistency and fairness to other approaches.
- Figure 6 — It would be interesting to see standard deviations for these bars.  I wonder how sensitive the results are to different initializations / different MCAR masks.  The authors mention that they are “significantly outperforming” other methods, but I think standard deviations are needed to justify this somewhat bold claim.  It would also be interesting to see how the percent missing from the data affects the results (currently it is fixed at 30%)?
- For conditional sampling (Algorithm 5), is there a reason why you need to generate step-by-step?  During training, it seems the model learns to predict all masked columns from unmasked ones in one step during training.  Do you get better performance by generating step-by-step at test time?

---

> ### Author Response · Authors · 2024-12-01
> **Response to Reviewer QPGs (1 of 4)**
>
> ### Nitpicks
> > Nitpick: The authors sometimes use overly promotional language, e.g. line 71 — “through two ingenious design features”, line 84 — “TABDAR offers several unparalleled advantages” — I recommend against the use of overly promotional words such as “ingenious”, “unparalleled advantages” in the paper.
>
> > Nitpick: Eq. (5) — Define $x^{<i}$. For a reader familiar with autoregressive modeling notation, it is obvious what this means, but for others it may not be. E.g. it would be good to clarify what this means for $i=1$.
>
> Thank you for your criticism. Regarding the first issue, in the revised paper, we have removed such adjectives. Regarding the second issue, in the revised paper, we have added a detailed explanation of the symbol $x^{<i}$ below Equation 1: $\mathbf{x}^{< i} = \lbrace x^1,x^2, \cdots, x^{i-1} \rbrace$.
>
> ### Clarify that in the training phase TabDAR is not an autoregressive model.
>
> > One point that was confusing to me at first was that the model is not trained as an autoregressive model -- it is trained as a masked language model. If I understand Eq. (8) / Algorithm 3 correctly, you are only masking out a particular set of columns at each training step. Thus, you are not really learning an autoregressive distribution, this is more like a conditional distribution like masked language modeling in which you predict p(masked | unmasked). I understand that you are using the learned conditional distribution in an autoregressive way at test time/generation time -- it would be good to clarify this point in the text.
>
> Thank you for your reminder. You are correct - during the training phase, TabDAR's masked language modeling is the same as BERT-like models. The difference is that BERT-like models aim to predict the exact values that were masked, while we aim to model their conditional distribution (mainly for continuous variables).
>
> In the generation phase, it is a genuine autoregressive generative model. In the revised paper, we have clearly stated this point in Section 1, line 83.
>
> ### Q1: Why using a diffusion loss for continuous columns.
>
> > For continuous columns, why use a diffusion loss? You could also use z to parameterize any simple continuous distribution, e.g. a Gaussian. Given that the diffusion process can be computationally expensive, do you think that it is a necessary component of the method? I understand that you do an ablation in which you compare against discretizing the continuous variable, but this also seems like an unnatural approach compared to simply passing it through a continuous distribution.
>
> Thank you for your insightful question. You're right, another ablation study should be using $z^i$ to parameterize a simple distribution with an analytical form.
>
> We did not consider using a simple Gaussian distribution at the time of submission because we felt that the conditional distribution of a continuous column should be relatively complex and difficult to capture with a simple Gaussian distribution. Since Diffusion models have already proven their powerful ability to model complex distributions, we directly adopted the Diffusion model.
>
> Your question reminded us that we could parameterize $p(x^i | z^i)$ as a Gaussian distribution, where both the expectation $\mu_i (z^i)$ and log standard deviation $\log \sigma(z_i)$ are output scalars from neural networks taking $z^i$ as input. However, we found that since $\mu_i (z^i)$ and $\log \sigma(z_i)$ are likely different for each data example, we don't have enough samples to calculate what the sample-based variance is for a specific $\log \sigma(z_i)$. This means we cannot optimize $\log \sigma(z_i)$. As for the expectation $\mu_i (z^i)$, this becomes equivalent to a regression model that regresses $x^i$ with $z^i$ as input, essentially degrading into a masked token prediction task.
>
> The following table shows the sample quality results obtained using the above method. (extension of Table 4, averaged results on all datasets)
>
> | Variants | Margin  | Joint |
> | ------ | --------- | ------ |
> | w/o both   | 14.82% | 8.94% |
> | w/o Diffusion loss   | 12.35% | 6.11% |
> | w/o random order   | 1.83% | 2.05% |
> | with simple Gaussian param.| 4.49% | 7.12% |
> |TabDAR|  **1.21%** | **1.80%** |
>
>
> These results indicate that using the above TabDAR variant makes it difficult to obtain sample quality that matches that of the Diffusion model.

---

> ### Author Response · Authors · 2024-12-01
> **Response to Reviewer QPGs (2 of 4)**
>
> ### Q2: How to distinguish between a continuous value being equal to 0 and the masked value
>
> > In your embedding/masking, how do you distinguish between a continuous value being equal to 0 and masking out that value? To me, it seems that both would lead to the same result after embedding the data. Is there a particular reason you chose to use zero for missing values instead of an embedding for a [mask] token, as typical in masked language models?
>
> Thank you for the reviewer's question. Our model distinguishes between inputs that are 0 and masked inputs. This is because the mask operation (Equation 8) occurs after adding the column-specific learnable positional encoding. For inputs that are 0, the column-specific positional embedding will be added to the input. However, for masked columns, the values will still be forced to 0 even after adding the positional embedding.
>
> We greatly appreciate your suggestion about using learnable [mask] token embeddings. We did experiment with this approach but observed that it provided limited improvements to the model's performance.
>
>
>
> ### Q3: Details of the diffusion model
>
> > Details on the diffusion aspect are generally missing. Is the denoising network the same for different continuous columns or is the denoising network column-specific? How many steps of diffusion are generally used?
>
> We apologize for not providing detailed explanations about our Diffusion model. We use one unique diffusion model for all continuous columns. We apologize for not explicitly stating this in the main text. In Equation (7), we implicitly indicated this by using the same denoising neural network notation $\varepsilon_{\theta}$ for all column indices $i$. For Diffusion steps, we use a fixed NFE = 50. These details have been updated in Section 5.1 of the revised paper.

---

> ### Author Response · Authors · 2024-12-01
> **Response to Reviewer QPGs (3 of 4)**
>
> ### Q4: Standard deviation of the results in Table 2
>
> > Table 2 — since some of these values are quite close, could you provide some standard deviations? Also, if there are ties with your method in the table, you should probably also bold these for consistency and fairness to other approaches.
>
> Thank you for your question. These methods show quite stable performance on the Machine Learning Efficiency task, with very small standard deviations. Due to space limitations, we couldn't add the standard deviations to Table 2. We show the complete results in the table below (mean and standard deviation are obtained via 20 random trials).
>
> | Methods | California  | Letter |  Car | Nursery | Adult | Default | Shoppers | Magic  | News | Beijing |
> | ------ | --------- | ------ | --------- | ------ |--------- | ------ |--------- | ------ |--------- | ------ |
> | Real | 0.999 $\pm$ 0.000 | 0.989 $\pm$ 0.001 | 0.999 $\pm$ 0.000 | 1.000 $\pm$ 0.000 | 0.927 $\pm$ 0.000 |  0.770 $\pm$ 0.004 | 0.926 $\pm$ 0.001 | 0.946 $\pm$ 0.001 | 0.842 $\pm$ 0.003 | 0.423 $\pm$ 0.002 |
> | TVAE | 0.986 $\pm$ 0.002 | 0.989 $\pm$ 0.002 | 0.746 $\pm$ 0.006 | 0.939 $\pm$ 0.008 | 0.846 $\pm$ 0.004 | 0.744 $\pm$ 0.005 | 0.898 $\pm$ 0.006 | 0.912 $\pm$ 0.004 | 0.979 $\pm$ 0.014 | 1.010 $\pm$ 0.016 |
> | GOGGLE | - | - | - | - | 0.778 $\pm$ 0.011 | 0.584 $\pm$ 0.005 | 0.658 $\pm$ 0.049 | 0.654 $\pm$ 0.021 |  0.877 $\pm$ 0.003 | 1.09 $\pm$ 0.025 |
> | CTGAN | 0.925 $\pm$ 0.003 | 0.729 $\pm$ 0.026 | 0.899 $\pm$ 0.011 | **1.000 $\pm$ 0.000** | 0.874 $\pm$ 0.002 | 0.736 $\pm$ 0.004 | 0.868 $\pm$ 0.008 | 0.874 $\pm$ 0.006 | 0.845 $\pm$ 0.015 | 1.065 $\pm$ 0.018 |
> | GReaT | 0.996 $\pm$ 0.001 | 0.983 $\pm$ 0.002 | 0.979 $\pm$ 0.002 | 0.999 $\pm$ 0.000 | 0.913 $\pm$ 0.003 | 0.755 $\pm$ 0.005 | 0.902 $\pm$ 0.006 | 0.888 $\pm$ 0.003 | - | 0.653 $\pm$ 0.013 |
> | STaSy | **0.997 $\pm$ 0.000** | 0.990 $\pm$ 0.001 | 0.927 $\pm$ 0.005 | 0.982 $\pm$ 0.003 | 0.903 $\pm$ 0.002 | 0.749 $\pm$ 0.006 | 0.909 $\pm$ 0.004 | 0.923 $\pm$ 0.004 | 0.933 $\pm$ 0.005 | 0.672 $\pm$ 0.016 |
> | CoDi | 0.981 $\pm$ 0.003 | **0.998 $\pm$ 0.000** | 0.995 $\pm$ 0.001 | 1.000 $\pm$ 0.000 | 0.829 $\pm$ 0.005 | 0.497 $\pm$ 0.006 | 0.855 $\pm$ 0.006 | 0.930 $\pm$ 0.004 | 0.999 $\pm$ 0.003 | 0.750 $\pm$ 0.014 |
> | TabDDPM | 0.992 $\pm$ 0.001 | 0.513 $\pm$ 0.026 | 0.995 $\pm$ 0.000 | **1.000 $\pm$ 0.000** | 0.911 $\pm$ 0.002 | 0.763 $\pm$ 0.004 | 0.915 $\pm$ 0.005 | 0.933 $\pm$ 0.003  | - | 2.665 $\pm$ 0.84 |
> | TabSyn | 0.993 $\pm$ 0.001 | 0.990 $\pm$ 0.002 | 0.971 $\pm$ 0.002 | 0.997 $\pm$ 0.001 |0.904 $\pm$ 0.002 | **0.764 $\pm$ 0.003** |  0.913 $\pm$ 0.003 | 0.93 $\pm$ 0.002 | 0.862 $\pm$ 0.025 | 0.669 $\pm$ 0.0011 |
> | DP-TBART | 0.993 $\pm$ 0.001 | 0.985 $\pm$ 0.003 | 0.990 $\pm$ 0.002 | 0.917 $\pm$ 0.002 | **0.918 $\pm$ 0.002** | 0.717 $\pm$ 0.004 | 0.896 $\pm$ 0.004 | 0.924 $\pm$ 0.002 | 0.896 $\pm$ 0.019 | 0.676 $\pm$ 0.008 |
>  | Tab-MT | 0.988 $\pm$ 0.003 |0.985 $\pm$ 0.004 | 0.981 $\pm$ 0.005 | **1.000 $\pm$ 0.000** | 0.873 $\pm$ 0.009 | 0.714 $\pm$ 0.007 | 0.912 $\pm$ 0.004 |0.822 $\pm$ 0.004 | 1.002$\pm$ 0.045 | 2.09 $\pm$ 0.19 |
> | TabDAR | 0.994 $\pm$ 0.001| 0.994 $\pm$ 0.001 | **0.996 $\pm$ 0.001** | **1.000 $\pm$ 0.000** | 0.904 $\pm$ 0.002 | **0.764 $\pm$ 0.003** | **0.916 $\pm$ 0.002** | **0.935 $\pm$ 0.002** | **0.856 $\pm$ 0.011** | **0.579 $\pm$ 0.005** |

---

> ### Author Response · Authors · 2024-12-01
> **Response to Reviewer QPGs (4 of 4)**
>
> ### Q5: Standard deviation of the bars
> > Figure 6 — It would be interesting to see standard deviations for these bars. I wonder how sensitive the results are to different initializations / different MCAR masks. The authors mention that they are “significantly outperforming” other methods, but I think standard deviations are needed to justify this somewhat bold claim. It would also be interesting to see how the percent missing from the data affects the results (currently it is fixed at 30%)?
>
> Thank you for your question. In the following table, we present the results with standard deviation (we will add and error bar to Figure 6 in the revised version)
>
> | MAE | TabDAR  | Remasker | MOT | GRAPE  | KNN |
> | ------ | --------- | ------ | --------- | ------  |  ------ |
> | Letter | 0.1978 $\pm$ 0.0015 | 0.2272 $\pm$ 0.0026 | 0.3495 $\pm$ 0.0012  | 0.3084 $\pm$ 0.002 | 0.3754 $\pm$ 0 |
> | Adult  | 0.4842 $\pm$ 0.0006 | 0.4679 $\pm$ 0.0022 | 0.4814 $\pm$ 0.0006 | 0.5684 $\pm$ 0.0028 | 0.5369 $\pm$ 0 |
> | Default | 0.2109 $\pm$ 0.0005 | 0.3403 $\pm$ 0.0018 | 0.2814 $\pm$ 0.0045 | 0.5251 $\pm$ 0.0047 | 0.2693 $\pm$ 0|
> | Shoppers | 0.3084 $\pm$ 0.0036 | 0.3245 $\pm$ 0.0016 | 0.3314 $\pm$ 0.0042 | 0.4204 $\pm$ 0.0050| 0.3689 $\pm$  0 |
>
>
> | RMSE | TabDAR  | Remasker | MOT | GRAPE  | KNN |
> | ------ | --------- | ------ | ------ | ------  |  ------ |
> | Letter | 0.3198 $\pm$ 0.0023| 0.4460 $\pm$ 0.0033| 0.5806 $\pm$ 0.0048| 0.5011 $\pm$ 0.0056| 0.5654 $\pm$ 0|
> | Adult  | 0.9190 $\pm$ 0.0011| 0.8852 $\pm$ 0.0042| 0.9167 $\pm$ 0.0062| 0.9684 $\pm$ 0.0065| 0.9269 $\pm$ 0|
> | Default | 0.5825 $\pm$ 0.0018| 0.7694 $\pm$ 0.0048| 0.6854 $\pm$ 0.0112| 0.9012 $\pm$ 0.0094| 0.6993 $\pm$ 0|
> | Shoppers | 0.6463 $\pm$ 0.0045| 0.6658 $\pm$ 0.0067 | 0.6863 $\pm$ 0.0174| 0.7837 $\pm$ 0.0152 | 0.6670 $\pm$ 0 |
>
>
> ### Q6: Reason for generating step-by-step (autoregressively)
>
> > For conditional sampling (Algorithm 5), is there a reason why you need to generate step-by-step? During training, it seems the model learns to predict all masked columns from unmasked ones in one step during training. Do you get better performance by generating step-by-step at test time?
>
>
> Yes, our experimental results show that step-by-step generation can achieve better generation quality than generating all columns at once.
>
> We can consider an extreme case, which is generating all columns at once in the unconditional generation scenario. In this case, our model is no longer an autoregressive generative model, but rather similar to a Diffusion Transformer. At this point, the generated data should come from $\prod_{i=1}^D p(x^i)$, which is the product of a series of marginal distributions, and it does not equal the joint distribution of the data $p(x^1, x^2, \cdots, x^D)$. And token-by-token autoregressive generation can ensure that we are sampling from the joint distribution.

---

### Public Comment · ~Calvin_McCarter1 · 2024-11-17
**Related work for autoregressive tabular modeling**

It's worth noting that gradient-boosted decision tree-based methods have previously shown performance exceeding neural network-based methods on this problem. ForestFlow [1] showed state-of-the-art performance for tabular generation, and UnmaskingTrees [2] showed state-of-the-art performance for tabular imputation. The latter work is of particular relevance, because it trains per-feature conditional models to autoregressively generate data, and avoids naive quantization via recursive hierarchical partitioning. It's also worth noting that while the paper claims "these models typically default to a left-to-right column sequence" (line 068), both TabMT [3] and UnmaskingTrees randomize over all possible column orderings.

[1] Jolicoeur-Martineau, Alexia, Kilian Fatras, and Tal Kachman. "Generating and imputing tabular data via diffusion and flow-based gradient-boosted trees." International Conference on Artificial Intelligence and Statistics. PMLR, 2024.

[2] McCarter, Calvin. "Unmasking Trees for Tabular Data." arXiv preprint arXiv:2407.05593 (2024).

[3] Gulati, Manbir, and Paul Roysdon. "TabMT: Generating tabular data with masked transformers." Advances in Neural Information Processing Systems 36 (2024).

---

> ### Author Response · Authors · 2024-11-17
> **Thanks for the reference**
>
> Hi Calvin.
>
> Thank you for mentioning the related works. We would like to clarify some points regarding your feedback：
>
> 1. [1] This can be viewed as replacing the denoising function (e.g., simple MLP) in the diffusion model with a Gradient boosting tree, while essentially it is still a Diffusion (Flow-matching) model. Since it did not compare with other advanced tabular data generation models (such as Tabsyn) in the experiments, we are not fully convinced of the view that "gradient-boosted decision tree-based methods have previously shown performance exceeding neural network-based methods on this problem". We agree that it would be beneficial to conduct an empirical comparison between our method and this approach.
>
> 2. We were not aware of your recent work [2] when writing this paper. We will read this work in detail and discuss its content in our revised version.
>
> 3. We acknowledge that TabMT [3] obtained different column orders through random shuffling. The sentence you cited, "these models typically default to a left-to-right column sequence (Castellon et al., 2023)" explicitly refers to another work, Dp-tbart, rather than Tab-MT. Moreover, the contribution of this paper is not in proposing the concept of random shuffling, but rather in simulating arbitrary sampling orders during the training phase through random masking.
>
> Thank you again for your attention to our work.

---

> > ### Public Comment · ~Calvin_McCarter1 · 2024-11-28
> > **Re TabMT**
> >
> > Actually, TabMT does not perform random shuffling. Rather, like your work, TabMT simulates random shuffling by marginalizing over random shuffles, to obtain a procedure that involves random masking -- see Eq (1) in TabMT.

---

> ### Author Response · Authors · 2024-12-01
>
> Thank you for your response. You are right, our method performs a similar mask prediction strategy. This is similar to TabMT and your Unmasking Trees. The main difference between our TabDAR and TabMT and your Unmasking Trees lies in how we learn the conditional distribution of each column. We will cite your recent work in the revised version.

---

### Meta-Review · Area_Chair_jeeq · 2024-12-20

**Metareview:**

The paper proposes a generative model for tabular data called Diffusion-nested Autoregressive model (TabDAR), which combines masked bi-directional attention transformers with diffusion models to handle mixed numerical and categorical data and supports arbitrary generation orders. The reviewers acknowledge that the paper tackles an important problem, proposes an interesting solution, and has extensive experiments. However, reviewers raised concerns regarding presentation issues, such as inconsistent notations and limited novelty compared to prior methods like TabSyn, which also exhibit similar empirical performance. Some reviewers also questioned the empirical discrepancies between reported baseline results and prior literature and found the experimental gains marginal on certain tasks, suggesting that benchmarks may not sufficiently differentiate methods. In the rebuttal, the authors addressed many technical and presentation issues. However, some conceptual critiques, such as novelty concerns, remain partially addressed. While the paper presents extensive experimental results, the marginal empirical gains, conceptual weaknesses, and mixed reviews on novelty mean that I recommend the paper be rejected.

**Additional Comments On Reviewer Discussion:**

The authors addressed many of the concerns raised by the reviewers and there was not post-rebuttal.

---

### Decision · Program_Chairs · 2025-01-22

Reject